# Counterfactual Explanations in Sequential Decision Making Under Uncertainty

**Stratis Tsirtsis**
MPI-SWS
stsirtsis@mpi-sws.org

**Abir De**
IIT Bombay
abir@cse.iitb.ac.in

**Manuel Gomez-Rodriguez**
MPI-SWS
manuelgr@mpi-sws.org

## Abstract

Methods to find counterfactual explanations have predominantly focused on one-step decision making processes. In this work, we initiate the development of methods to find counterfactual explanations for decision making processes in which multiple, dependent actions are taken sequentially over time. We start by formally characterizing a sequence of actions and states using finite horizon Markov decision processes and the Gumbel-Max structural causal model. Building upon this characterization, we formally state the problem of finding counterfactual explanations for sequential decision making processes. In our problem formulation, the counterfactual explanation specifies an alternative sequence of actions differing in at most $k$ actions from the observed sequence that could have led the observed process realization to a better outcome. Then, we introduce a polynomial time algorithm based on dynamic programming to build a counterfactual policy that is guaranteed to always provide the optimal counterfactual explanation on every possible realization of the counterfactual environment dynamics. We validate our algorithm using both synthetic and real data from cognitive behavioral therapy and show that the counterfactual explanations our algorithm finds can provide valuable insights to enhance sequential decision making under uncertainty.

## 1 Introduction

In recent years, there has been a rising interest on the potential of machine learning models to assist and enhance decision making in high-stakes applications such as justice, education and healthcare [1–3]. In this context, machine learning models and algorithms have been used not only to predict the outcome of a decision making process from a set of observable features but also to find what would have to change in (some of) the features for the outcome of a specific process realization to change. For example, in loan decisions, a bank may use machine learning both to estimate the probability that a customer repays a loan and to find by how much a customer may need to reduce her credit card debt to increase the probability of repayment over a given threshold. Here, our focus is on the latter, which has been often referred to as counterfactual explanations.

The rapidly growing literature on counterfactual explanations has predominantly focused on one-step decision making processes as the one described above [4–6]. In such settings, the probability that an outcome occurs is typically estimated using a supervised learning model and finding counterfactual explanations reduces to a search problem across the space of features and model predictions [7–11]. Moreover, it has been argued that, to obtain counterfactual explanations that are actionable and have the predicted effect on the outcome, one should favor causal models [12–14]. In this work, our goal is instead to find counterfactual explanations for decision making processes in which multiple, dependent actions are taken sequentially over time. In this setting, the (final) outcome of the process depends on the overall sequence of actions and the counterfactual explanation specifies an alternative sequence of actions differing in at most $k$ actions from the observed sequence that could have led the process realization to a better outcome. For example, in medical treatment, assume a physician takes a

35th Conference on Neural Information Processing Systems (NeurIPS 2021).

sequence of actions to treat a patient but the patient's prognosis does not improve, then a counterfactual explanation would help the physician understand how a small number of actions taken differently could have improved the patient's prognosis. However, since there is (typically) uncertainty on the counterfactual dynamics of the environment, there may be a different counterfactual explanation that is optimal under each possible realization of the counterfactual dynamics. Moreover, since, in many realistic scenarios, a decision maker needs to decide among a small number of actions on the basis of a few observable covariates, which are often discretized into (percentile) ranges, in this work, we focus on decision making processes where the state and action spaces are discrete and low-dimensional.

The work most closely related to ours, which lies within the area of machine learning for healthcare, has achieved early success at using machine learning to enhance sequential (clinical) decisions [15–18]. However, rather than finding counterfactual explanations, this line of work has predominantly focused on using reinforcement learning to design better treatment policies. A notable exception is by Oberst and Sontag [18], which introduces an off-policy evaluation procedure for highlighting specific realizations of a sequential decision making process where a policy trained using reinforcement learning would have led to a substantially different outcome. While our work builds upon their modeling framework, we do not focus on off-policy evaluation but, instead, on the generation of (better) alternative action sequences as a means of explanation for the process's outcome. Therefore, we see their contributions as complementary to ours.

More broadly, our work is not the first to use counterfactual reasoning in the context of sequential decision making in the machine learning literature [19, 20]. However, previous work has used counterfactual reasoning either to learn optimal policies using limited observational data [19] or to explain the action choices of a reinforcement learning agent [20]. In contrast, our work uses a causal model of the environment to compute alternative action sequences, close to the observed one, which, in retrospect, could have improved the outcome of the decision making process. Refer to Appendix A for a discussion of further related work.

**Our contributions.** We start by formally characterizing a sequence of (discrete) actions and (discrete) states using finite horizon Markov decision processes (MDPs). Here, we model the transition probabilities between a pair of states, given an action, using the Gumbel-Max structural causal model [18]. This model has been shown to have a desirable counterfactual stability property and, given a sequence of actions and states, it allows us to reliably estimate the counterfactual outcome under an alternative sequence of actions. Building upon this characterization, we formally state the problem of finding the counterfactual explanation for an observed realization of a sequential decision making process as a constrained search problem over the set of alternative sequences of actions differing in at most $k$ actions from the observed sequence. Then, we present a polynomial time algorithm based on dynamic programming that finds a counterfactual policy that is guaranteed to always provide the optimal counterfactual explanation on every possible realization of the counterfactual transition probability induced by the observed process realization. Finally, we validate our algorithm using both synthetic and real data from cognitive behavioral therapy and show that the counterfactual explanations can provide valuable insights to enhance sequential decision making under uncertainty[1].

## 2 Characterizing Sequential Decision Making using Causal Markov Decision Processes

Our starting point is the following stylized setting, which resembles a variety of real-world sequential decision making processes. At each time step $t \in \{0, \ldots, T-1\}$, the decision making process is characterized by a state $s_t \in \mathcal{S}$, where $\mathcal{S}$ is a space of $n$ states, an action $a_t \in \mathcal{A}$, where $\mathcal{A}$ is a space of $m$ actions, and a reward $R(s_t, a_t) \in \mathbb{R}$. Moreover, given a realization of a decision making process $\tau = \{(s_t, a_t)\}_{t=0}^{T-1}$, the outcome of the decision making process $o(\tau) = \sum_t R(s_t, a_t)$ is given by the sum of the rewards.

Given the above setting, we characterize the relationship between actions, states and outcomes using finite horizon Markov decision processes (MDPs). More specifically, we consider an MDP $\mathcal{M} = (\mathcal{S}, \mathcal{A}, P, R, T)$, where $\mathcal{S}$ is the state space, $\mathcal{A}$ is the set of actions, $P$ denotes the transition probability $P(S_{t+1} = s_{t+1} \mid S_t = s_t, A_t = a_t)$, $R$ denotes the immediate reward $R(s_t, a_t)$, and $T$ is the time horizon. While this characterization is helpful to make predictions about future states and design action policies [21], it is not sufficient to make counterfactual predictions, *e.g.*, given a realization of a decision making process $\tau = \{(s_t, a_t)\}_{t=0}^{T-1}$, we cannot know what would have

---

[1]Our code is accessible at https://github.com/Networks-Learning/counterfactual-explanations-mdp.

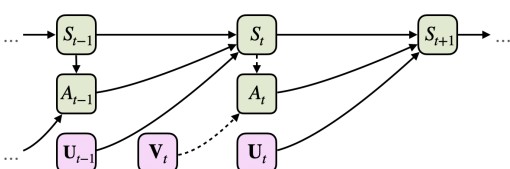

Figure 1: Structural causal model $\mathcal{C}$ for a Markov decision process $\mathcal{M}$. Green boxes represent endogenous random variables and pink boxes represent exogenous noise variables. The value of each endogenous variable is given by a function of the values of its ancestors in the structural causal model, as defined by Eq. 1. The value of each exogenous noise variable is sampled independently from a given distribution. An intervention $do(A_t = a)$ breaks the dependence of the variable $A_t$ from its ancestors (highlighted by dotted lines) and sets its value to $a$. After observing an event $S_{t+1} = s_{t+1}, S_t = s_t, A_t = a_t$, a counterfactual prediction can be thought of as the result of an intervention $do(A_t = a)$ in a modified SCM where $U_t$ takes values $\boldsymbol{u}_t$ from a posterior distribution with support such that $s_{t+1} = g_S(s_t, a_t, \boldsymbol{u}_t)$.

happened if, instead of taking action $a_t$ at time $t$, we had taken action $a' \neq a_t$. To be able to overcome this limitation, we will now augment the above characterization using a particular class of structural causal model (SCM) [22, 23] with desirable properties.

Let $\mathcal{C}$ be a structural causal model defined by the assignments

$$S_{t+1} = g_S(S_t, A_t, \boldsymbol{U}_t) \quad \text{and} \quad A_t = g_A(S_t, \boldsymbol{V}_t), \tag{1}$$

where $\boldsymbol{U}_t$ and $\boldsymbol{V}_t$ are $n$- and $m$-dimensional independent noise variables, respectively, and $g_S$ and $g_A$ are two given functions, and let $P^{\mathcal{C}}$ be the distribution entailed by $\mathcal{C}$. Then, as argued by Buesing et al. [19], we can always find a distribution for the noise variables and a function $g_S$ so that the transition probability of the MDP of interest is given by the following interventional distribution over the SCM $\mathcal{C}$:

$$P(S_{t+1} = s_{t+1} \mid S_t = s_t, A_t = a_t) = P^{\mathcal{C} \,;\, do(A_t = a_t)}(S_{t+1} = s_{t+1} \mid S_t = s_t) \tag{2}$$

where $do(A_t = a_t)$ denotes a (hard) intervention in which the second assignment in Eq. 1 is replaced by the value $a_t$.

Under this view, given an observed realization of a decision making process $\tau = \{(s_t, a_t)\}_{t=0}^{T-1}$, we can compute the posterior distribution $P^{\mathcal{C} \mid S_t = s_t, S_{t+1} = s_{t+1}, A_t = a_t}(\boldsymbol{U}_t)$ of each noise variable $\boldsymbol{U}_t$. Building on the conditional density function of this posterior distribution, which we denote as $f_{\boldsymbol{U}_t}^{\mathcal{C} \mid S_t = s_t, S_{t+1} = s_{t+1}, A_t = a_t}(\boldsymbol{u})$, we can define a (non-stationary) counterfactual transition probability

$$P_{\tau, t}(S_{t+1} = s' \mid S_t = s, A_t = a) = P^{\mathcal{C} \mid S_t = s_t, S_{t+1} = s_{t+1}, A_t = a_t \,;\, do(A_t = a)}(S_{t+1} = s' \mid S_t = s)$$

$$= \int_{\mathbb{R}^n} P^{\mathcal{C} \mid S_t = s_t, S_{t+1} = s_{t+1}, A_t = a_t \,;\, do(A_t = a)}(S_{t+1} = s' \mid S_t = s, \mathbf{U}_t = \boldsymbol{u})$$

$$\times f_{\mathbf{U}_t}^{\mathcal{C} \mid S_t = s_t, S_{t+1} = s_{t+1}, A_t = a_t \,;\, do(A_t = a)}(\boldsymbol{u})d\boldsymbol{u}$$

$$\overset{(a)}{=} \int_{\mathbb{R}^n} \mathbf{1}[s' = g_S(s, a, \boldsymbol{u})] \cdot f_{\mathbf{U}_t}^{\mathcal{C} \mid S_t = s_t, S_{t+1} = s_{t+1}, A_t = a_t}(\boldsymbol{u})d\boldsymbol{u}$$

$$= \mathbb{E}_{\mathbf{U}_t \mid S_t = s_t, S_{t+1} = s_{t+1}, A_t = a_t}[\mathbf{1}[s' = g_S(s, a, \mathbf{U_t})]], \tag{3}$$

where, in (a), we drop the $do(\cdot)$ because $\mathbf{U}_t$ and $A_t$ are independent in the modified SCM. Importantly, the above counterfactual transition probability allows us to make counterfactual predictions, *e.g.*, given that, at time $t$ the state was $s_t$ and, at time $t + 1$, the process transitioned to state $s_{t+1}$ after taking action $a_t$, what would have been the probability of transitioning to state $s'$ after taking action $a \neq a_t$ if the state had been $s$ at time $t$. Refer to Figure 1 for a visual description of our model and the notion of counterfactual predictions.

However, for state variables taking discrete values, the posterior distribution of the noise may be non-identifiable without further assumptions, as argued by Oberst and Sontag [18]. This is because there may be several noise distributions and functions $g_S$ which give interventional distributions consistent with the MDP's transition probabilities but result in different counterfactual transition distributions. To avoid these non-identifiability issues, we follow Oberst and Sontag and restrict our attention to the class of Gumbel-Max SCMs, *i.e.*,

$$S_{t+1} = g_S(S_t, A_t, \boldsymbol{U}_t) := \operatorname*{argmax}_{s \in \mathcal{S}} \{\log P(S_{t+1} = s \mid S_t, A_t) + U_{t,s}\}, \tag{4}$$

where $U_{t,s} \sim \text{Gumbel}(0, 1)$ and $P(\cdot \mid S_t, A_t)$ is the transition probability of the MDP. More specifically, this class of SCMs has been shown to satisfy a desirable counterfactual stability property, which goes intuitively as follows. Assume that, at time $t$, the process transitioned from state $s_t$ to state $s_{t+1}$ after taking action $a_t$. Then, in a counterfactual scenario, it is unlikely that, at time $t$, the process would transition from a state $s$ to a state $s'$ other than $s_{t+1}$—the factual one—unless choosing an action $a$ that decreases the relative chances of $s_{t+1}$ compared to the other states. More formally, for a model satisfying counterfactual stability, given $\tau = \{(s_t, a_t)\}_{t=0}^{T-1}$, for all $s, s'$ with $s' \neq s_{t+1}$ such that

$$\frac{P(S_{t+1} = s_{t+1} \mid S_t = s, A_t = a)}{P(S_{t+1} = s_{t+1} \mid S_t = s_t, A_t = a_t)} \geq \frac{P(S_{t+1} = s' \mid S_t = s, A_t = a)}{P(S_{t+1} = s' \mid S_t = s_t, A_t = a_t)},$$

it holds that $P_{\tau,t}(S_{t+1} = s' \mid S_t = s, A_t = a) = 0$. In practice, in addition to solving the non-identifiability issues, the use of Gumbel-Max SCMs allows for an efficient procedure to sample from the corresponding noise posterior distribution $P^{\mathcal{C} \mid S_t = s_t, S_{t+1} = s_{t+1}, A_t = a_t}(\mathbf{U}_t)$, described elsewhere [18, 24], and given a set of $d$ samples from the noise posterior distribution, we can compute an unbiased finite sample Monte-Carlo estimator for the counterfactual transition probability, as defined in Eq. 3, as follows:

$$P_{\tau,t}(S_{t+1} = s' \mid S_t = s, A_t = a) \approx \frac{1}{d} \sum_{j \in [d]} \mathbf{1}[s' = g_S(s, a, \boldsymbol{u}_j)] \tag{5}$$

## 3 Counterfactual Explanations in Markov Decision Processes

Inspired by previous work on counterfactual explanations in supervised learning [7, 8], we focus on counterfactual outcomes that could have occurred if the alternative action sequence was "close" to the observed one. However, since in our setting, there is uncertainty on the counterfactual dynamics of the environment, we will look for a non-stationary counterfactual policy $\pi$ that, under every possible realization of the counterfactual transition probability defined in Eq. 3, is guaranteed to provide the optimal alternative sequence of actions differing in at most $k$ actions from the observed one.

More specifically, let $\tau = \{(s_t, a_t)\}_{t=0}^{T-1}$ be an observed realization of a decision making process characterized by a Markov decision process (MDP) $\mathcal{M} = (\mathcal{S}, \mathcal{A}, P, R, T)$ with a transition probability defined via a Gumbel-Max structural causal model (SCM), as described in Section 2. Then, to characterize the effect that any alternative action sequence would have had on the outcome of the above process realization, we start by building a non-stationary counterfactual MDP $\mathcal{M}_\tau = (\mathcal{S}^+, \mathcal{A}, P_\tau^+, R^+, T)$. Here, $\mathcal{S}^+ = \mathcal{S} \times \{0, \ldots, T-1\}$ is an enhanced state space such that each $s^+ \in \mathcal{S}^+$ corresponds to a pair $(s, l)$ indicating that the original decision making process would have been at state $s \in \mathcal{S}$ had already taken $l$ actions differently from the observed sequence. Following this definition, $R^+$ denotes the reward function which we define as $r^+((s, l), a) = R(s, a)$ for any $(s, l) \in \mathcal{S}^+$ and $a \in \mathcal{A}$, *i.e.*, the counterfactual rewards remain independent of the number of modifications in the action sequence. Lastly, let $P_\tau$ be the counterfactual transition probability, as defined by Eq. 3. Then, the transition probability $P_\tau^+$ for the enhanced state space is defined as:

$$P_{\tau,t}^+ \left( S_{t+1}^+ = (s', l') \mid S_t^+ = (s, l), A_t = a \right) = \begin{cases} P_{\tau,t} \left( S_{t+1} = s' \mid S_t = s, A_t = a \right) \\ \quad \text{if } (a = a_t \wedge l' = l) \vee (a \neq a_t \wedge l' = l+1) \\ 0 \quad \text{otherwise,} \end{cases}$$

where note that the dynamics of the original states $s$ are equivalent both under $P_{\tau,t}^+$ and $P_{\tau,t}$, however, under $P_{\tau,t}^+$, we also keep track of the number of actions differing from the observed actions. Now, let $\pi : \mathcal{S}^+ \times \{0, \ldots, T-1\} \to \mathcal{A}$ be a policy that deterministically decides about the counterfactual action $a_t'$ that should have been taken if the process's enhanced state had been $s_t^+ = (s_t', l_t)$, *i.e.*, the counterfactual state at time $t$ was $s_t'$ after performing $l_t$ action changes. Then, given such a counterfactual policy $\pi$, we can compute the corresponding average counterfactual outcome as follows:

$$\bar{o}_\pi(\tau) := \mathbb{E}_{\tau' \sim P_\tau^+ \mid s_0^+ = (s_0, 0)} \left[ \sum_{t=0}^{T-1} r^+((s_t', l_t), a_t') \right] \tag{6}$$

where $\tau' = \{((s_t', l_t), a_t')\}_{t=0}^{T-1}$ is a realization of the non-stationary counterfactual MDP $\mathcal{M}_\tau$ with $a_t' = \pi((s_t', l_t), t)$ and the expectation is taken over all the realizations induced by the transition probability $P_\tau^+$ and the policy $\pi$. Here, note that, if $\pi((s_t, 0), t) = a_t$ for all $t \in \{0, \ldots, T-1\}$, then $\bar{o}_\pi(\tau) = o(\tau)$ matches the outcome of the observed realization.

**ALGORITHM 1:** It samples a counterfactual explanation from the counterfactual policy $\pi$

---

**Input**: counterfactual policy $\pi$, horizon $T$, counterfactual transition probability $P_\tau$, reward function $R$, initial state $s_0$.

$s'_0 \leftarrow s_0$
$l_0 \leftarrow 0$
$\text{reward} \leftarrow 0$
**for** $t = 0, \dots, T-1$ **do**
    $a'_t \leftarrow \pi((s'_t, l_t), t)$
    $\text{reward} \leftarrow \text{reward} + R(s'_t, a'_t)$
    **if** $t \neq T-1$ **then**
        $s'_{t+1} \sim P_{\tau,t}(S_{t+1} \mid S_t = s'_t, A_t = a'_t)$
        **if** $a'_t \neq a_t$ **then**
            $l_{t+1} \leftarrow l_t + 1$
        **else**
            $l_{t+1} \leftarrow l_t$
        **end**
    **end**
**end**
$\tau' \leftarrow \{((s'_t, l_t), a'_t)\}_{t=0}^{T-1}$
$o(\tau') \leftarrow \text{reward}$
**Return** $\tau', o(\tau')$

---

Then, our goal is to find the optimal counterfactual policy $\pi_\tau^*$ that maximizes the counterfactual outcome subject to a constraint on the number of counterfactual actions that can differ from the observed ones, *i.e.*,

$$\underset{\pi}{\text{maximize}} \quad \bar{o}_\pi(\tau) \qquad \text{subject to} \quad \sum_{t=0}^{T-1} \mathbf{1}[a_t \neq a'_t] \leq k \quad \forall \tau' \sim P_\tau^+ \tag{7}$$

where $a'_1, \dots, a'_{T-1}$ is one realization of counterfactual actions and $a_1, \dots, a_{T-1}$ are the observed actions. The constraint guarantees that any counterfactual action sequence induced by the counterfactual transition probability $P_\tau^+$ and the counterfactual policy $\pi$ can differ in at most $k$ actions from the observed sequence. Finally, once we have found the optimal policy $\pi_\tau^*$, we can sample a counterfactual realization of the process and the optimal counterfactual explanation using Algorithm 1.

## 4 Finding Optimal Counterfactual Explanations via Dynamic Programming

To solve the problem defined by Eq. 7, we break the problem into several smaller sub-problems. Here, the key idea is to compute the counterfactual policy values that lead to the optimal counterfactual outcome recursively by expanding the expectation in Eq. 6 for one time step.

We start by computing the highest average cumulative reward $h(s, r, c)$ that one could have achieved in the last $r$ steps of the decision making process, starting from state $S_{T-r} = s$, if at most $c$ actions had been different to the observed ones in those last steps. For $c > 0$, we proceed recursively:

$$h(s, r, c) = \max \left( R(s, a_{T-r}) + \sum_{s' \in \mathcal{S}} P_{\tau, T-r}(s' \mid s, a_{T-r}) h(s', r-1, c), \right.$$

$$\left. \max_{a \in \mathcal{A} \,:\, a \neq a_{T-r}} \left[ R(s, a) + \sum_{s' \in \mathcal{S}} P_{\tau, T-r}(s' \mid s, a) h(s', r-1, c-1) \right] \right), \tag{8}$$

and, for $c = 0$, we trivially have that:

$$h(s, r, 0) = R(s, a_{T-r}) + \sum_{s' \in \mathcal{S}} P_{\tau, T-r}(s' \mid s, a_{T-r}) h(s', r-1, 0), \tag{9}$$

with $s \in \mathcal{S}$, $r \in \{1, \dots, T\}$, $c \in \{1, \dots, k\}$, and $h(s, 0, c) = 0$ for all $s$ and $c$. In Eq. 8, the first parameter of the outer maximization corresponds to the case where, at time $T - r$, the observed action $a_{T-r}$ is taken and the second parameter corresponds to the case where, instead of the observed action, the best alternative action is taken.

**ALGORITHM 2:** It returns the optimal counterfactual policy and its average counterfactual outcome

---

**Input**: States $\mathcal{S}$, actions $\mathcal{A}$, realization $\tau$, horizon $T$, counterfactual transition probability $P_\tau$, reward function $R$, constraint $k$.

**Initialize**: $h(s, r, c) \leftarrow 0, \quad s \in \mathcal{S}, r = 0, \ldots, T, c = 0, \ldots, k$.

**for** $r = 1, \ldots, T$ **do**

    **for** $s \in \mathcal{S}$ **do**

        $h(s, r, 0) \leftarrow R(s, a_{T-r})$ ;       /* Base cases: No action changes left (Eq. 9) */

        **for** $s' \in \mathcal{S}$ **do**

            $h(s, r, 0) \leftarrow h(s, r, 0) + P_{\tau, T-r}(s' \,|\, s, a_{T-r}) h(s', r-1, 0)$

        **end**

        $\pi_\tau^*((s, k), T - r) \leftarrow a_{T-r}$ ;          /* Take the observed action */

    **end**

**end**

**for** $r = 1, \ldots, T$ **do**

    **for** $c = 1, \ldots, k$ **do**

        **for** $s \in \mathcal{S}$ **do**

            reward $\leftarrow R(s, a_{T-r})$ ;       /* Compute the first part of Eq. 8 */

            **for** $s' \in \mathcal{S}$ **do**

                reward $\leftarrow$ reward $+ P_{\tau, T-r}(s' \,|\, s, a_{T-r}) h(s', r-1, c)$

            **end**

            best_reward $\leftarrow$ reward

            best_action $\leftarrow a_{T-r}$

            **for** $a \in \mathcal{A} \setminus \{a_{T-r}\}$ **do**

                reward_alt $\leftarrow R(s, a)$ ;       /* Compute the second part of Eq. 8 */

                **for** $s' \in \mathcal{S}$ **do**

                    reward_alt $\leftarrow$ reward_alt $+ P_{\tau, T-r}(s' \,|\, s, a) h(s', r-1, c-1)$

                **end**

                **if** *reward_alt > best_reward* **then**

                    best_reward $\leftarrow$ reward_alt

                    best_action $\leftarrow a$

                **end**

            **end**

            $h(s, r, c) \leftarrow$ best_reward

            $\pi_\tau^*((s, k - c), T - r) \leftarrow$ best_action ;    /* Take the action maximizing Eq. 8 */

        **end**

    **end**

**end**

**Return** $\pi_\tau^*, h(s_0, T, k)$

---

By definition, we can easily conclude that $h(s_0, T, k)$ is the average counterfactual outcome of the optimal counterfactual policy $\pi_\tau^*$, *i.e.*, the objective value of the solution to the optimization problem defined by Eq. 7, and we can recover the optimal counterfactual policy $\pi_\tau^*$ by keeping track of the action chosen at each recursive step in Eq. 8 and 9. The overall procedure, summarized by Algorithm 2, uses dynamic programming—it first computes the values $h(s, 1, c)$ for all $s$ and $c$ and then proceeds with the rest of computations in a bottom-up fashion—and has complexity $\mathcal{O}(n^2 m T k)$. Finally, we have the following proposition (proven by induction in Appendix B):

**Proposition 1** *The counterfactual policy $\pi_\tau^*$ returned by Algorithm 2 is the solution to the optimization problem defined by Eq. 7.*

## 5 Experiments on Synthetic Data

In this section, we evaluate Algorithm 2 on realizations of a synthetic decision making process[2]. To this end, we first look into the average outcome improvement that could have been achieved if at most $k$ actions had been different to the observed ones in every realization, as dictated by the optimal counterfactual policy. Then, we investigate to what extent the level of uncertainty of the decision making process influences the average counterfactual outcome achieved by the optimal counterfactual policy as well as the number of distinct counterfactual explanations it provides.

---

[2] All experiments were performed on a machine equipped with 48 Intel(R) Xeon(R) 3.00GHz CPU cores and 1.5TB memory.

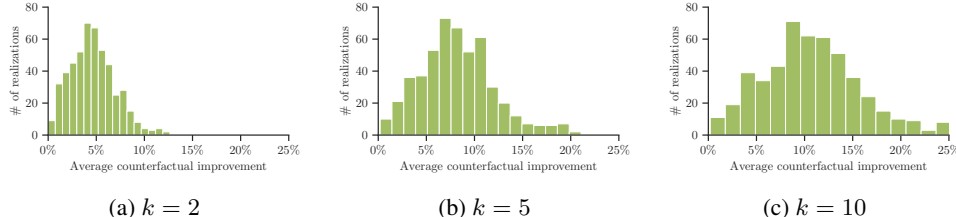

(a) $k = 2$            (b) $k = 5$            (c) $k = 10$

Figure 2: Empirical distribution of the relative difference between the average counterfactual outcome $\bar{o}_{\pi^*_\tau}(\tau)$ achieved by the optimal counterfactual policy $\pi^*_\tau$ and the observed outcome $o(\tau)$, *i.e.*, $\mathbb{P}[(\bar{o}_{\pi^*_\tau}(\tau) - o(\tau))/o(\tau)]$, in a synthetic decision making process. In all panels, we set $n = 20$, $m = 10$, $\alpha = 0.4$, $d = 1{,}000$ and estimate the distributions using 500 realizations from 10 different instances of the decision making process (50 realizations per instance), each with different $w_s$.

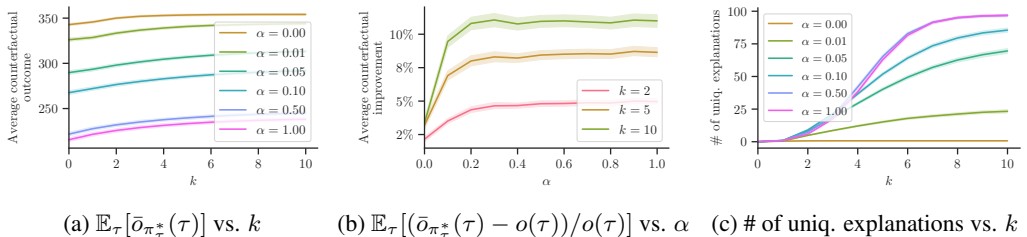

(a) $\mathbb{E}_\tau[\bar{o}_{\pi^*_\tau}(\tau)]$ vs. $k$     (b) $\mathbb{E}_\tau[(\bar{o}_{\pi^*_\tau}(\tau) - o(\tau))/o(\tau)]$ vs. $\alpha$     (c) # of uniq. explanations vs. $k$

Figure 3: Influence that the level of uncertainty of a synthetic decision making process has on the average counterfactual outcome $\bar{o}_{\pi^*_\tau}(\tau)$ achieved by the optimal counterfactual policy $\pi^*_\tau$ as well as on the number of distinct counterfactual explanations $\pi^*_\tau$ provides. In all panels, we set $n = 20$, $m = 10$ and $d = 1{,}000$ and, in each experiment, use 500 realizations from 10 different instances of the decision making process (50 realizations per instance), each with different $w_s$. In panel (c), for each realization, we sample 100 counterfactual realizations and compute the average number of unique explanations across realizations. Shaded regions correspond to 95% confidence intervals.

**Experimental setup.** We characterize the synthetic decision making process using an MDP with states $\mathcal{S} = \{0, \ldots, n-1\}$ and actions $\mathcal{A} = \{0, \ldots, m-1\}$, where $n = 20$ and $m = 10$, and time horizon $T = 20$. For each state $s$ and action $a$, we set the immediate reward equal to $R(s, a) = s$, *i.e.*, the higher the state, the higher the reward. To set the values of the transition probabilities $P(S_{t+1} \mid S_t = s_t, A_t = a_t)$, we proceed as follows. First we pick one $s^\star \in \mathcal{S}$ uniformly at random and we set a weight $w_{s^\star} = 1$ and then, for the remaining states $s \in \mathcal{S} \setminus s^\star$, we sample weights $w_s \sim U[0, \alpha]$, where $\alpha \leq 1$. Next, for all $s \in \mathcal{S}$, we set $P(S_{t+1} = s \mid S_t = s_t, A_t = a_t) = w_s / \sum_{s' \in \mathcal{S}} w_{s'}$. It is easy to check that, for each state-action pair $(s_t, a_t)$ at time $t$, $s_{t+1} = s^\star$ is most likely to be observed in the next timestep $t + 1$. The parameter $\alpha$ controls the level of uncertainty.

Then, we compute the optimal policy that maximizes the average outcome of the decision making process by solving Bellman's equation using dynamic programming [21] and use this policy to sample the (observed) realizations as follows. For each realization, we start from a random initial state $s_0 \in \mathcal{S}$ and, at each time step $t$, we pick the action indicated by the optimal policy with probability 0.95 and a different action uniformly at random with probability 0.05. This leads to action sequences that are slightly suboptimal in terms of average outcome[3]. Finally, to compute the counterfactual transition probability $P_{\tau,t}$ for each observed realization $\tau$, we follow the procedure described in Section 2 with $d = 1{,}000$ samples for each noise posterior distribution[4].

**Results.** We first measure to what extent the counterfactual explanations provided by the optimal counterfactual policy $\pi^*_\tau$ would have improved the outcome of the decision making process. To this end, for each observed realization $\tau$, we compute the relative difference between the average

---

[3]We introduce this randomization to emulate slightly suboptimal (human) policies. However, note that an observed action sequence may be suboptimal in retrospect even if it was picked using an optimal policy.

[4]For the estimation of the counterfactual transition probabilities $P_\tau$ of a single realization $\tau$, we report an average execution time of 62 seconds.

optimal counterfactual outcome and the observed outcome, *i.e.*, $(\bar{o}_{\pi^*_\tau}(\tau) - o(\tau))/o(\tau)$. Figure 2 summarizes the results for different values of $k$. We find that the relative difference between the average counterfactual outcome and the observed outcome is always positive, *i.e.*, the sequence of actions specified by the counterfactual explanations would have led the process realization to a better outcome in expectation. However, this may not come as a surprise given that the counterfactual policy $\pi^*_\tau$ is optimal, as shown in Proposition 1. In this context, note that, in the worst case, the counterfactual policy $\pi^*_\tau$ would trivially repeat the observed action sequence, leading to an average counterfactual outcome equal to the observed outcome with probability 1. Moreover, we observe that, as the sequences of actions specified by the counterfactual explanations differ more from the observed actions (*i.e.*, $k$ increases), the improvement in terms of expected outcome increases.

Next, we investigate how the level of uncertainty $\alpha$ of the decision making process influences the average counterfactual outcome achieved by the optimal counterfactual policy $\pi^*_\tau$ as well as the number of distinct counterfactual explanations $\pi^*_\tau$ provides. Figure 3 summarizes the results, which reveal several interesting insights. As the level of uncertainty $\alpha$ increases, the average counterfactual outcome decreases, as shown in panel (a), however, the relative difference with respect to the observed outcome increases, as shown in panel (b). This suggest that, under high level of uncertainty, the counterfactual explanations may be more valuable to a decision maker who aims to improve her actions over time. However, in this context, we also find that, under high levels of uncertainty, the number of distinct counterfactual explanations increases rapidly with $k$. As a result, it may be preferable to use relatively low values of $k$ to be able to effectively show the counterfactual explanations to a decision maker in practice.

## 6  Experiments on Real Data

In this section, we evaluate Algorithm 2 using real patient data from a series of cognitive behavioral therapy sessions. Similarly as in Section 5, we first measure the average outcome improvement that could have been achieved if at most $k$ actions had been different to the observed ones in every therapy session, as dictated by the optimal counterfactual policy. Then, we look into individual therapy sessions and showcase how Algorithm 2, together with Algorithm 1, can be used to highlight specific patients and actions of interest for closer inspection[5]. Appendix D contains additional experiments benchmarking the optimal counterfactual policy against several baselines[6].

**Experimental setup.** We use anonymized data from a clinical trial comparing the efficacy of hypnotherapy and cognitive behavioral therapy [25] for the treatment of patients with mild to moderate symptoms of major depression[7]. In our experiments, we use data from the 77 patients who received manualized cognitive behavioral therapy, which is one of the gold standards in depression treatment. Among these patients, we discard four of them because they attended less than 10 sessions. Each patient attended up to 20 weekly therapy sessions and, for each session, the dataset contains the theme of discussion, chosen by the therapist from a pre-defined set of themes (*e.g.*, psychoeducation, behavioural activation, cognitive restructuring techniques). Additionally, a severity score is included, based on a standardized questionnaire [27], filled by the patient at each session, which assesses the severity of depressive symptoms. For more details about the severity score and the pre-defined set of discussion themes refer to Appendix C.

To derive the counterfactual transition probability for each patient, we start by creating an MDP with $n = 5$ states and $m = 11$ actions. Each state $s \in \mathcal{S} = \{0, \ldots, 4\}$ corresponds to a severity score, where small numbers represent lower severity, and each action $a \in \mathcal{A}$ corresponds to a theme from the pre-defined list of themes that the therapists discussed during the sessions. Moreover, each realization of the MDP corresponds to the therapy sessions of a single patient ordered in chronological order and time horizon $T \in \{10, \ldots, 20\}$ is the number of therapy sessions per patient. Here, we denote the set of realizations for all patients as $\mathcal{T}$.

In addition, to estimate the values of the transition probabilities, we proceed as follows. For every state-action pair $(s_i, a)$, we assume a $n$-dimensional Dirichlet$(\alpha_{i1}, \ldots, \alpha_{i1})$ prior on the probabilities $p_{j\,|\,i,a} = P(S_{t+1} = s_j \,|\, S_t = s_i, A_t = a)$, where $\alpha_{ij} = 1$ if $j \in \{i - 1, i, i + 1\}$

---

[5]Our results should be interpreted in the context of our modeling assumptions and they do not suggest the existence of medical malpractice.

[6]We do not evaluate our algorithm against prior work on counterfactual explanations for one-step decision making processes since the settings are not directly comparable.

[7]All participants gave written informed consent and the study protocol was peer-reviewed [26].

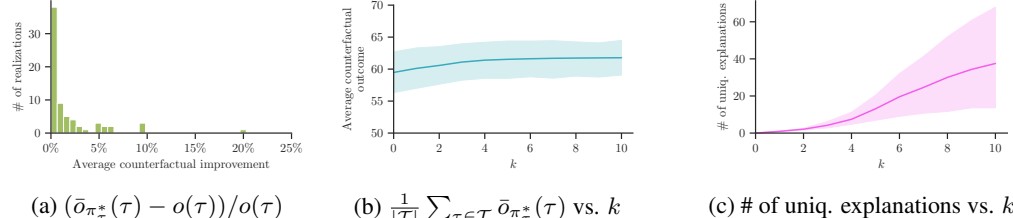

(a) $(\bar{o}_{\pi_\tau^*}(\tau) - o(\tau))/o(\tau)$  (b) $\frac{1}{|\mathcal{T}|} \sum_{\tau \in \mathcal{T}} \bar{o}_{\pi_\tau^*}(\tau)$ vs. $k$  (c) # of uniq. explanations vs. $k$

Figure 4: Performance achieved by the optimal counterfactual policy $\pi_\tau^*$ in a series of real manualized cognitive behavioral therapy sessions $\mathcal{T}$, where each realization $\tau \in \mathcal{T}$ includes all the sessions of an individual patient sorted in chronological order. Panel (a) shows the distribution of the relative difference between the average counterfactual outcome $\bar{o}_{\pi_\tau^*}(\tau)$ achieved by $\pi_\tau^*$ and the observed outcome $o(\tau)$, i.e., $(\bar{o}_{\pi_\tau^*}(\tau) - o(\tau))/o(\tau)$, for $k = 3$ (refer to Appendix E for additional results under other $k$ values). Panels (b) and (c) show the average counterfactual outcome $\bar{o}_{\pi_\tau^*}(\tau)$ achieved by $\pi_\tau^*$ and the average number of unique counterfactual explanations provided by each $\pi_\tau^*$, averaged across patients, against the number of actions $k$ differing from the observed ones. In panel (c), for each realization, the average number of unique counterfactual explanations provided by $\pi_\tau^*$ is estimated using 1,000 counterfactual realizations. In all panels, we set $d = 1,000$ and use data from 73 patients. Shaded regions correspond to 95% confidence intervals.

and $\alpha_{ij} = 0.01$ otherwise. Then, if we observe $c_j$ transitions from state $s_i$ to each state $s_j$ after action $a$ in the patients' therapy sessions $\mathcal{T}$, we have that the posterior of the probabilities $p_{j\,|\,i,a}$ is a Dirichlet$(\alpha_{i1} + c_1, \ldots, \alpha_{in} + c_n)$. Finally, to estimate the value of the transition probabilities $P(S_{t+1}\,|\,S_t = s_i, A_t = a)$, we take the average of 100,000 samples from the posterior probability $p_{j\,|\,i,a}$. This procedure sets the value of the transition probabilities proportionally to the number of times they appeared in the data, however, it ensures that all transition probability values are non zero and transitions between adjacent severity levels are much more likely to happen. Moreover, we set the immediate reward for a pair of state and action $(s, a)$ equal to $R(s, a) = 5 - s \in \{1, \ldots, 5\}$, i.e., the lower the patient's severity level, the higher the reward. Here, if some state-action pair $(s, a)$ is never observed in the data, we set its immediate reward to $R(s, a) = -\infty$. This ensures that those state-action pairs never appear in a realization induced by the optimal counterfactual policy. Finally, to compute the counterfactual transition probability $P_{\tau,t}$ for each realization $\tau \in \mathcal{T}$, we follow the procedure described in Section 2 with $d = 1,000$ samples for each noise posterior distribution.

**Results.** We first measure to what extent the counterfactual explanations provided by the optimal counterfactual policy $\pi_\tau^*$ would have improved each patient's severity of depressive symptoms over time. To this end, for each observed realization $\tau \in \mathcal{T}$ corresponding to each patient, we compute the same quality metrics as in experiments on synthetic data in Section 5. Figure 4 summarizes the results. Panel (a) reveals that, for most patients, the improvement in terms of relative difference between the average optimal counterfactual outcome $\bar{o}_{\pi_\tau^*}(\tau)$ and the observed outcome $o(\tau)$ is rather modest. Moreover, panel (b) also shows that the absolute average optimal counterfactual outcome $\bar{o}_{\pi_\tau^*}(\tau)$, averaged across patients, does not increase significantly even if one allows for more changes $k$ in the sequence of observed actions. These findings suggest that, in retrospect, the choice of themes by most therapists in the sessions was almost optimal. That being said, for 20% of the patients, the average counterfactual outcome improves a $\geq 3$ % over the observed outcome and, as we will later discuss, there exist individual counterfactual realizations in which the counterfactual outcome improves much more than 3%. In that context, it is also important to note that, as shown in panel (c), the growth in the number of unique counterfactual explanations with respect to $k$ is weaker than the growth found in the experiments with synthetic data and, for $k \leq 4$, the number of unique counterfactual explanations is smaller than 10. This latter finding suggests that, in practice, it may be possible to effectively show, or summarize, the optimal counterfactual explanations, a possibility that we investigate next.

We focus on a patient for whom the average counterfactual outcome $\bar{o}_{\pi_\tau^*}(\tau)$ achieved by the optimal policy $\pi_\tau^*$ with $k = 3$ improves 9.5% over the observed outcome $o(\tau)$. Then, using the policy $\pi_\tau^*$, also with $k = 3$, and the counterfactual transition probability $P_\tau$, we sample multiple counterfactual explanations $\tau'$ using Algorithm 1 and look at each counterfactual outcome $o(\tau')$. Figure 5(a) summarizes the results, which show that, in most of these counterfactual realizations, the counterfactual outcome is greater than the observed outcome—if at most $k$ actions had been different to the observed

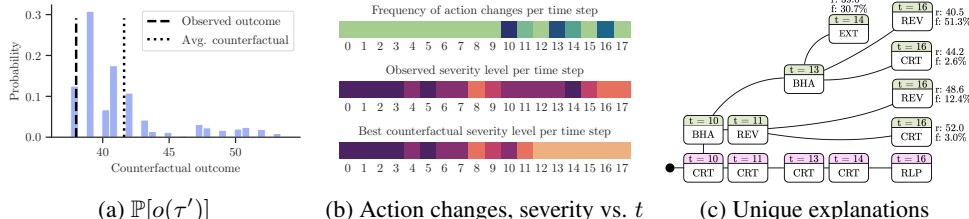

(a) $\mathbb{P}[o(\tau')]$  (b) Action changes, severity vs. $t$  (c) Unique explanations

Figure 5: Insights on the counterfactual explanations provided by the optimal counterfactual policy $\pi_\tau^*$ for one real patient who received manualized cognitive behavioral therapy. Panel (a) shows the distribution of the counterfactual outcomes $o(\tau')$ for the counterfactual realizations $\tau'$ induced by $\pi_\tau^*$ and $P_\tau$. Panel (b) shows, for each time step, how frequently a counterfactual explanation changes the observed action as well as the observed severity level and the severity level in the counterfactual realization with the highest counterfactual outcome. Here, darker colors correspond to higher frequencies and higher severities. Panel (c) shows the action changes in the unique counterfactual explanations (green) provided by $\pi_\tau^*$ along with the mean of counterfactual outcomes (r) that each one achieves and how frequently (f) they appear across the counterfactual realizations. Here, the bottom row shows the observed actions that were changed by at least one of the counterfactual explanations. Refer to Appendix C for a definition of the actions (i.e., themes). In all panels, we set $d = 1,000$ and the results are estimated using $1,000$ counterfactual realizations.

ones, as dictated by the optimal policy, there is a high probability that the outcome would have improved. Next, we investigate to what extent there are specific time steps within the counterfactual realizations $\tau'$ where $\pi_\tau^*$ is more likely to suggest an action change. Figure 5(b) shows that, for the patient under study, there are indeed time steps that are overrepresented in the optimal counterfactual explanations, namely $t \in \{10, 13, 16\}$. Moreover, the first of these time steps ($t = 10$) is when the patient had started worsening their depression after an earlier period in which they showed signs of recovery. Remarkably, we find that, in the counterfactual realization $\tau'$ with the best counterfactual outcome, the worsening is mostly avoided. Finally, we look closer into the actual action changes suggested by the optimal counterfactual policy $\pi_\tau^*$. Figure 5(c) summarizes the results, which reveal that $\pi_\tau^*$ recommends replacing some of the sessions on cognitive restructuring techniques (CRT) by behavioral activation (BHA) consistently across counterfactual realizations $\tau'$, particularly at the start of the worsening period. We discussed this recommendation with one of the researchers on clinical psychology who co-authored Fuhr et al. [25] and she told us that, from a clinical perspective, such recommendation is sensible since, whenever the severity of depressive symptoms is high, it is very challenging to apply CRT and instead it is quite common to use BHA. Appendix F contains additional insights about other patients in the dataset.

## 7 Conclusions, Limitations and Future Work

We have initiated the study of counterfactual explanations in decision making processes in which multiple, dependent actions are taken sequentially over time. Building on a characterization of sequential decision making processes using MDPs and the Gumbel-Max SCM, we have developed a polynomial time algorithm to find optimal counterfactual explanations. Using synthetic and real data from cognitive behavioral therapy, we have shown that the counterfactual explanations our algorithm finds can provide valuable insights to enhance sequential decision making under uncertainty.

Our work opens up many interesting avenues for future work, which may solve some of its limitations. For example, we have considered sequential decision making processes with discrete states and actions satisfying the Markov property. Although this setting may fit a plethora of real-world applications, it would be interesting to extend our work to continuous states and actions and/or semi-Markov processes. Moreover, we experimentally validated our method using a single real dataset. It would be valuable to evaluate counterfactual explanations generated by our algorithm using additional datasets from other (medical) applications. In this context, it would be worth to consider applications in which the true transition probabilities are due to a machine learning algorithm. Finally, the usefulness of the counterfactual explanations given by our algorithm crucially depends on the particular structural causal model we have focused on. To make such explanations more practical, it would be important to consider alternative structural causal models and carry out a user study in which the counterfactual explanations are shared with the human experts (e.g., therapists) who took the observed actions.

**Acknowledgements.** We would like to thank Kristina Fuhr and Anil Batra for giving us access to the cognitive behavioral therapy data that made this work possible. Tsirtsis and Gomez-Rodriguez acknowledge support from the European Research Council (ERC) under the European Union's Horizon 2020 research and innovation programme (grant agreement No. 945719). De has been partially supported by a DST Inspire Faculty Award.

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
