# A  Further related work

Our work also builds upon further related work on interpretable machine learning and counterfactual inference. In terms of interpretable machine learning, in addition to the work on counterfactual explanations for one-step decision making processes discussed in Section 1 [4–6], there is also a popular line of work focused on feature-based explanations [28–30]. Feature-based explanations highlight the importance each feature has on a particular prediction by a model, typically through local approximation. In this context, one particular type of feature-based explanations that have been relatively popular in explaining the action choices of reinforcement learning agents in simple gaming environments (*e.g.*, Atari games) are those in the form of saliency maps [31–33]. In terms of counterfactual inference, the literature has a long history [34], however, it has primarily focused on estimating quantities related to the counterfactual distribution of interest such as, *e.g.*, the conditional average treatment effect (CATE). More broadly, one may also draw connections between our work and automated planning [35, 36] and off-policy evaluation in reinforcement learning [37–39].

# B  Proof of Proposition 1

Using induction, we will prove that the policy value $\pi_\tau((s, l), t')$ set by Algorithm 2 is optimal for every $s \in \mathcal{S}$, $l \in \{0, \ldots, k\}$, $t' \in \{0, \ldots, T-1\}$ in the sense that following this policy maximizes the average cumulative reward $h(s, r, c)$ that one could have achieved in the last $r = T - t'$ steps of the decision making process, starting from state $S_{T-r} = s$, if at most $c = k - l$ actions had been different to the observed ones in those last steps. Formally:

$$h(s, r, c) = \max_\pi \mathbb{E}_{\{((s_t', l_t), a_t')\}_{t=t'}^{T-1} \sim P_\tau^+ \mid S_{t'}^+ = (s,l)} \left[ \sum_{t=t'}^{T-1} r^+ \left( (s_t', l_t), a_t' \right) \right] \tag{10}$$

$$\text{subject to} \sum_{t=t'}^{T-1} \mathbf{1}[a_t \neq a_t'] \leq c \quad \forall \{((s_t', l_t), a_t')\}_{t=t'}^{T-1} \sim P_\tau^+ \tag{11}$$

Recall that, $a_1, \ldots, a_{T-1}$ are the observed actions and the counterfactual realizations $a_1', \ldots, a_{T-1}'$ are induced by the counterfactual transition probability $P_\tau^+$ and the policy $\pi$.

We start by proving the induction basis. Assume that a realization has reached a state $s_{T-1}^+ = (s, l)$ at time $T - 1$, one time step before the end of the process. If $c = 0$ (*i.e.*, $l = k$), following Equation 9, the algorithm will choose the observed action $\pi_\tau((s, l), t') = a_{T-1}$ and return an average cumulative reward $h(s, 1, 0) = R(s, a_{T-1}) + \sum_{s' \in \mathcal{S}} P_{\tau, T-1}(s' \mid s, a_{T-1}) h(s', 0, 0) = R(s, a_{T-1})$, where $h(s', 0, 0) = 0$ for all $s' \in \mathcal{S}$. Since no more action changes can be performed at this stage, this is the only feasible solution and therefore it is optimal.

If $c > 0$, since $h(s', 0, c) = h(s', 0, c - 1) = 0$ for all $s' \in \mathcal{S}$ it is easy to verify that Equation 8 reduces to $h(s, 1, c) = \max_{a \in \mathcal{A}} R(s, a)$ and $\pi_\tau((s, l), t') = \operatorname{argmax}_{a \in \mathcal{A}} R(s, a)$ is obviously the optimal choice for the last time step.

Now, we will prove that, for a counterfactual realization being at state $s_{t'}^+ = (s, l)$ at a time step $t' < T - 1$ (*i.e.*, $r = T - t'$, $c = k - l$), the maximum average cumulative reward $h(s, r, c)$ given by Algorithm 2 is optimal, under the inductive hypothesis that the values of $h(s', r', c')$ already computed for $r' < r$, $c' < c$ and all $s' \in \mathcal{S}$ are optimal. Assume that the algorithm returns an average cumulative reward $h(s, r, c)$ by choosing action $\pi_\tau((s, l), t') = a$ while the optimal solution gives an average cumulative reward $OPT_{s,r,c} > h(s, r, c)$ by choosing an action $a^* \neq a$. Here, by $\tau_{t'}' = \{((s_t', l_t), a_t')\}_{t=t'}^{T-1}$ we will denote realizations starting from time $t'$ with $a_t' = \pi_\tau((s_t', l_t), t)$ where $\pi_\tau$ is the policy given by Algorithm 2 and we will use $\tau_{t'}^*$ if the policy is optimal. Also, we will denote a possible next state at time $t' + 1$, after performing action $a$, as $(s', l')$ where $l' = l + 1$ if $a \neq a_t$, $l' = l$ otherwise and, $c' = k - l'$. Similarly, after performing action $a^*$, we will denote a possible next state as $(s', l^*)$ where $l^* = l + 1$ if $a^* \neq a_t$, $l^* = l$ otherwise and, $c^* = k - l^*$. Then, we get:

$$h(s, r, c) < OPT_{s,r,c}$$

$$\implies \mathbb{E}_{\tau_{t'}' \sim P_\tau^+ \mid S_{t'}^+ = (s,l)} \left[ \sum_{t=t'}^{T-1} r^+ \left( (s_t, l_t), a_t \right) \right] < \mathbb{E}_{\tau_{t'}^* \sim P_\tau^+ \mid S_{t'}^+ = (s,l)} \left[ \sum_{t=t'}^{T-1} r^+ \left( (s_t, l_t), a_t \right) \right]$$

and further we have that

$$\mathbb{E}_{\tau'_{t'}\sim P_\tau^+ \,|\, S_{t'}^+=(s,l)}\left[\sum_{t=t'}^{T-1} r^+\left((s_t,l_t),a_t\right)\right] < \mathbb{E}_{\tau_{t'}^*\sim P_\tau^+ \,|\, S_{t'}^+=(s,l)}\left[\sum_{t=t'}^{T-1} r^+\left((s_t,l_t),a_t\right)\right]$$

$$\overset{(a)}{\Longrightarrow} \sum_{s'} P_{\tau,T-r}(s'\,|\,s,a)\mathbb{E}_{\tau'_{t'+1}\sim P_\tau^+ \,|\, S_{t'+1}^+=(s',l')}\left[\sum_{t=t'}^{T-1} r^+\left((s_t,l_t),a_t\right)\right]$$

$$< \sum_{s'} P_{\tau,T-r}(s'\,|\,s,a^*)\mathbb{E}_{\tau_{t'+1}^*\sim P_\tau^+ \,|\, S_{t'+1}^+=(s',l^*)}\left[\sum_{t=t'}^{T-1} r^+\left((s_t,l_t),a_t\right)\right]$$

$$\Longrightarrow \sum_{s'} P_{\tau,T-r}(s'\,|\,s,a)\left[r^+\left((s,l),a\right) + \mathbb{E}_{\tau'_{t'+1}\sim P_\tau^+ \,|\, S_{t'+1}^+=(s',l')}\left[\sum_{t=t'+1}^{T-1} r^+\left((s_t,l_t),a_t\right)\right]\right]$$

$$< \sum_{s'} P_{\tau,T-r}(s'\,|\,s,a^*)\left[r^+\left((s,l),a^*\right) + \mathbb{E}_{\tau_{t'+1}^*\sim P_\tau^+ \,|\, S_{t'+1}^+=(s',l^*)}\left[\sum_{t=t'+1}^{T-1} r^+\left((s_t,l_t),a_t\right)\right]\right]$$

$$\overset{(b)}{\Longrightarrow} \sum_{s'} P_{\tau,T-r}(s'\,|\,s,a)R(s,a) + \sum_{s'} P_{\tau,T-r}(s'\,|\,s,a)h(s',r-1,c')$$

$$< \sum_{s'} P_{\tau,T-r}(s'\,|\,s,a^*)R(s,a^*) + \sum_{s'} P_{\tau,T-r}(s'\,|\,s,a^*)OPT_{s',r-1,c^*}$$

$$\overset{(c)}{\Longrightarrow} R(s,a) + \sum_{s'} P_{\tau,T-r}(s'\,|\,s,a)h(s',r-1,c')$$

$$< R(s,a^*) + \sum_{s'} P_{\tau,T-r}(s'\,|\,s,a^*)h(s',r-1,c^*),$$

where, in (a), we expand the expectation for one time step, in (b), we replace the average cumulative reward starting from time step $t'+1$ with $h(s',r-1,c')$ and $OPT_{s',r-1,c^*}$ for the policy of Algorithm 2 and the optimal one respectively and, in (c), we replace $OPT_{s',r-1,c^*}$ with $h(s',r-1,c^*)$ due to the inductive hypothesis.

It is easy to see that, it can either be $a^* = a_t$ with $c^* = c$ or $a^* \in \mathcal{A} \setminus a_t$ with $c^* = c - 1$. If $c = 0$, following Equation 9, the algorithm will choose the observed action (*i.e.*, $a = a_t$). This is the only feasible solution, since $a^* \neq a_t$ would give $c^* = -1$ and $l^* = k - c^* = k + 1$, which is not a valid state. Therefore, we get $a = a^* = a_t$, which is a a contradiction. If $c > 0$, because of the max operator in Equation 8, for the action $a$ chosen by Algorithm 2, it necessarily holds that:

$$R(s,a) + \sum_{s'} P_{\tau,T-r}(s'\,|\,s,a)h(s',r-1,c') \geq R(s,a^*) + \sum_{s'} P_{\tau,T-r}(s'\,|\,s,a^*)h(s',r-1,c^*),$$

which is clearly a contradiction.

Therefore, the average cumulative reward $h(s,r,c)$ computed by Algorithm 2 and its associated policy value $\pi_\tau((s,l),t')$ are optimal for every $s \in \mathcal{S}$, $l \in \{0,\ldots,k\}$, $t' \in \{0,\ldots,T-1\}$ and $h(s_0,T,k)$ is the solution to the optimization problem defined by Equation 7.

## C  Additional Details about the Cognitive Behavioral Therapy Dataset

Each patient's severity of depression is measured using the standardized questionnaire PHQ-9 [27], which consists of 9 questions regarding the frequency of depressive symptoms (*e.g.*, "Feeling tired or having little energy?") manifested over a period of two weeks. The patient has to answer each question by placing themselves on a scale ranging from 0 ("Not at all") to 3 ("Nearly every day"). The sum of those answers, ranging from 0 to 27, reflects the overall depression severity and it is usually discretized into five categories, corresponding to no depression $(0-4)$, mild depression $(5-9)$, moderate depression $(10-14)$, moderately severe depression $(15-19)$, severe depression $(20-27)$[8]. In our experiments, the states $\mathcal{S} = \{0,\ldots,4\}$ correspond to these five categories.

---

[8]The full version of the questionnaire can be found at `https://patient.info/doctor/patient-health-questionnaire-phq-9`.

Each session of cognitive behavioral therapy contains information about the topic of discussion between the patient and the therapist, among 24 pre-defined topics [25], with some of the topics having similar content. For example, there were 4 topics about "cognitive restructuring techniques" which, we observed that, some therapists merged and covered in 2 sessions. Here, we grouped the above topics into the following eleven broader themes:

- STR – First session: Introduction, discussing expectations, getting to know each other, discussing the current symptoms / problems, current life situation.
- BIO – Biography: A look at biography, family and social frame of reference, school and professional development, emotional development, partnerships, important turning points or crises.
- PSE – Psychoeducation: Discuss symptoms of depression, recognize and understand connections between feelings, thoughts and behavior (depression triangle) based on a situation analysis from the current / last episode, causes of depression, develop a disease model, explain the treatment approach in relation to the model.
- BHA – Behavioural activation: Focus on behaviour, discuss the vicious circle (depression spiral), discuss list of pleasant activities, attention to life balance, if necessary improve the daily structure, recognizing and eliminating obstacles and problems.
- REV – Review: Review of the last sessions, collection of strategies learned so far, find suitable strategies for typical situations, draft a personal strategy plan, plan further steps.
- CRT – Cognitive restructuring techniques: Discuss influence of thoughts on feelings and actions, identify thought patterns, discuss influence of automatic thoughts / basic assumptions, check the validity of automatic thoughts.
- INR – Interactional competence: Self-assessment of your own self-confidence, discuss current interpersonal issues and derive goals, carry out role plays, transfer into everyday life.
- THP – Re-evaluation of thought patterns: Review, evaluate and rename basic assumptions, schemes and general plans.
- RLP – Relapse prevention: Explain the risk of relapse, discuss early warning symptoms, recognize risk situations, develop suitable strategies.
- END – Closing session: Finding a good end to the therapy, looking back on the last 5-6 months, parting ritual.
- EXT – Extra material: Sleep disorders, problem-solving skills, brooding module "When thinking doesn't help", discuss the influence of rumination on mood and impairments in everyday life, progressive muscle relaxation.

In our experiments, the actions $\mathcal{A}$ correspond to these broader themes. However, since the themes STR and END appeared only in the first ($t = 0$) and last ($t = T - 1$) time steps of each realization, we kept them fixed and we did not allow these themes to be used as action changes during the time steps $t = \{1, \ldots, T - 2\}$.

## D  Performance Comparison with Baseline Policies

**Experimental setup.** In this section, we compare the average counterfactual outcome achieved by the optimal counterfactual policy, given by Algorithm 2, with that achieved by several baseline policies. To this end, we use the same experimental setup as in Section 6, however, instead of setting $R(s, a) = -\infty$ for every unobserved pair $(s, a)$, we set $R(s, a) = 5 - s \in \{1, \ldots, 5\}$, similarly as for the observed pairs. This is because, otherwise, we observed that there were always realizations under the baselines policies for which the counterfactual outcome was $-\infty$. In our experiments, we consider with the following baselines policies:

- **Random**: At each time step $t$, the policy chooses the next action $a^*$ uniformly at random if $l_t < k$ and it chooses $a^* = a_t$ otherwise.
- **Greedy**: At each time step $t$, being at state $(s'_t, l_t)$, the policy chooses the next action $a^*$ greedily, *i.e.*, if $l_t < k$, then

$$a^* = \underset{a \in \mathcal{A}}{\arg\max} \ R(s, a) + \sum_{s' \in \mathcal{S}} P_{\tau, t}(S_{t+1} = s' \,|\, S_t = s'_t, A_t = a) R(s', a'), \qquad (12)$$

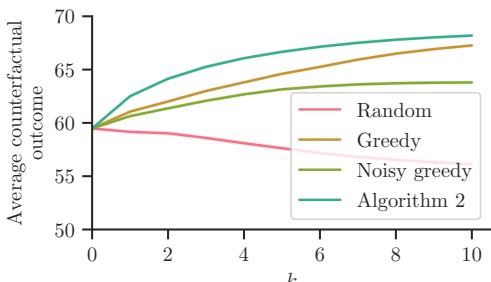

Figure 6: Performance achieved by the optimal counterfactual policy $\pi^*_\tau$ given by Algorithm 2 and the baseline policies in the same series of cognitive behavioral therapy sessions $\mathcal{T}$ introduced in Section 6. The plot shows the average counterfactual outcome $\frac{1}{\mathcal{T}}\sum_{\tau\in\mathcal{T}}\bar{o}_{\pi_\tau}(\tau)$ achieved by $\pi^*_\tau$ and the baseline policies, averaged over the set of observed realizations $\mathcal{T}$, against the number of actions $k$ differing from the observed ones. For each observed realization, the average counterfactual outcome is estimated using 1,000 counterfactual realizations. Here, we set $d = 1,000$ and use data from $|\mathcal{T}| = 73$ patients. Shaded regions correspond to 95% confidence intervals.

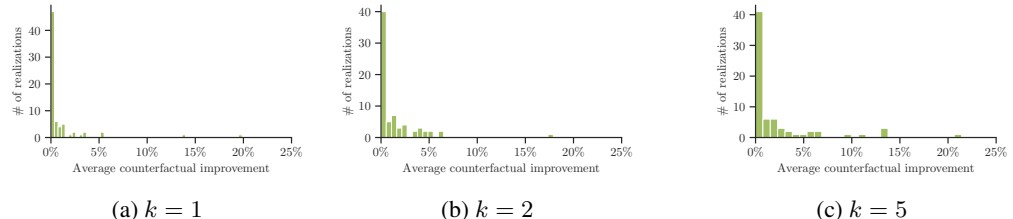

(a) $k = 1$          (b) $k = 2$          (c) $k = 5$

Figure 7: Empirical distribution of the relative difference between the average counterfactual outcome $\bar{o}_{\pi^*_\tau}(\tau)$ achieved by the optimal counterfactual policy $\pi^*_\tau$ and the observed outcome $o(\tau)$, *i.e.*, $\mathbb{P}[(\bar{o}_{\pi^*_\tau}(\tau) - o(\tau))/o(\tau)]$ for several values of $k$. Here, we consider the same series of cognitive behavioral therapy sessions $\mathcal{T}$ introduced in Section 6. In all panels, we set $d = 1,000$ and use data from 73 patients.

    and, if $l_t = k$, $a^* = a_t$.

- **Noisy greedy**: At each time step $t$, being at state $(s'_t, l_t)$, it chooses the next action $a^*$ as follows. If $l_t < k$, $a^*$ is given by Eq. 12 with probability 0.5 and $a^* = a_t$ otherwise. If $l_t = k$, $a^* = a_t$.

**Results.** Figure 6 shows the average counterfactual outcomes achieved by the optimal policy, as given by Algorithm 2, and the above baselines for different $k$ values. The results show that, as expected, the optimal policy outperforms all the baselines across the entire range of $k$ values and, moreover, the competitive advantage is greater for smaller $k$ values. In addition, we also find that the performance of the random baseline policy drops significantly as $k$ increases, since, as discussed in Section 6, the observed trajectories are close to optimal in retrospect and, differing from them causes the random policy to worsen the counterfactual outcome.

# E   Average counterfactual improvement for additional values of k

In this section, we use the same experimental setup as in Section 6 to measure to what extent the counterfactual explanations provided by the optimal counterfactual policy $\pi^*_\tau$ would have improved each patient's severity of depressive symptoms, under other values of $k$. To this end, for each observed realization $\tau \in \mathcal{T}$ corresponding to each patient, we compute the relative difference between the average optimal counterfactual outcome and the observed outcome, *i.e.*, $(\bar{o}_{\pi^*_\tau}(\tau) - o(\tau))/o(\tau)$.

Figure 7 summarizes the results, which show that, similarly as in Section 6, for most patients, the improvement in terms of relative difference between the average optimal counterfactual outcome $\bar{o}_{\pi^*_\tau}(\tau)$ and the observed outcome $o(\tau)$ is modest. That being said, we observe that, as the sequences of actions specified by the counterfactual explanations differ more from the observed actions (*i.e.*, $k$ increases), the improvement in terms of expected outcome presents a slight increase.

# F   Insights About Individual Patients

In this section, we provide insights about additional patients in the dataset. For each of these additional patients, we follow the same procedure in Section 6, *i.e.*, we use Algorithm 1 with the policy $\pi_\tau^*$, with $k = 3$, to sample multiple counterfactual explanations $\tau'$ and look at the corresponding counterfactual outcomes $o(\tau')$. Figure 8 summarizes the results, where each row corresponds to a different patient. The results reveal several interesting insights. For most of the patients, all of the counterfactual realizations lead to counterfactual outcomes greater or equal than the observed outcome (left column), however, the difference between the average counterfactual outcome and the observed outcome is relatively small. Notable exceptions are a few patients for whom there is a small probability that the counterfactual outcome is worse than the observed one (top row) as well as patients for whom the difference between the average counterfactual outcome and the observed outcome is high (bottom row). Additionally, we also find that the actual action changes suggested by the optimal counterfactual policies $\pi_\tau^*$ are typically concentrated in a few time steps across counterfactual realizations (right column), usually at the beginning or the end of the realizations.

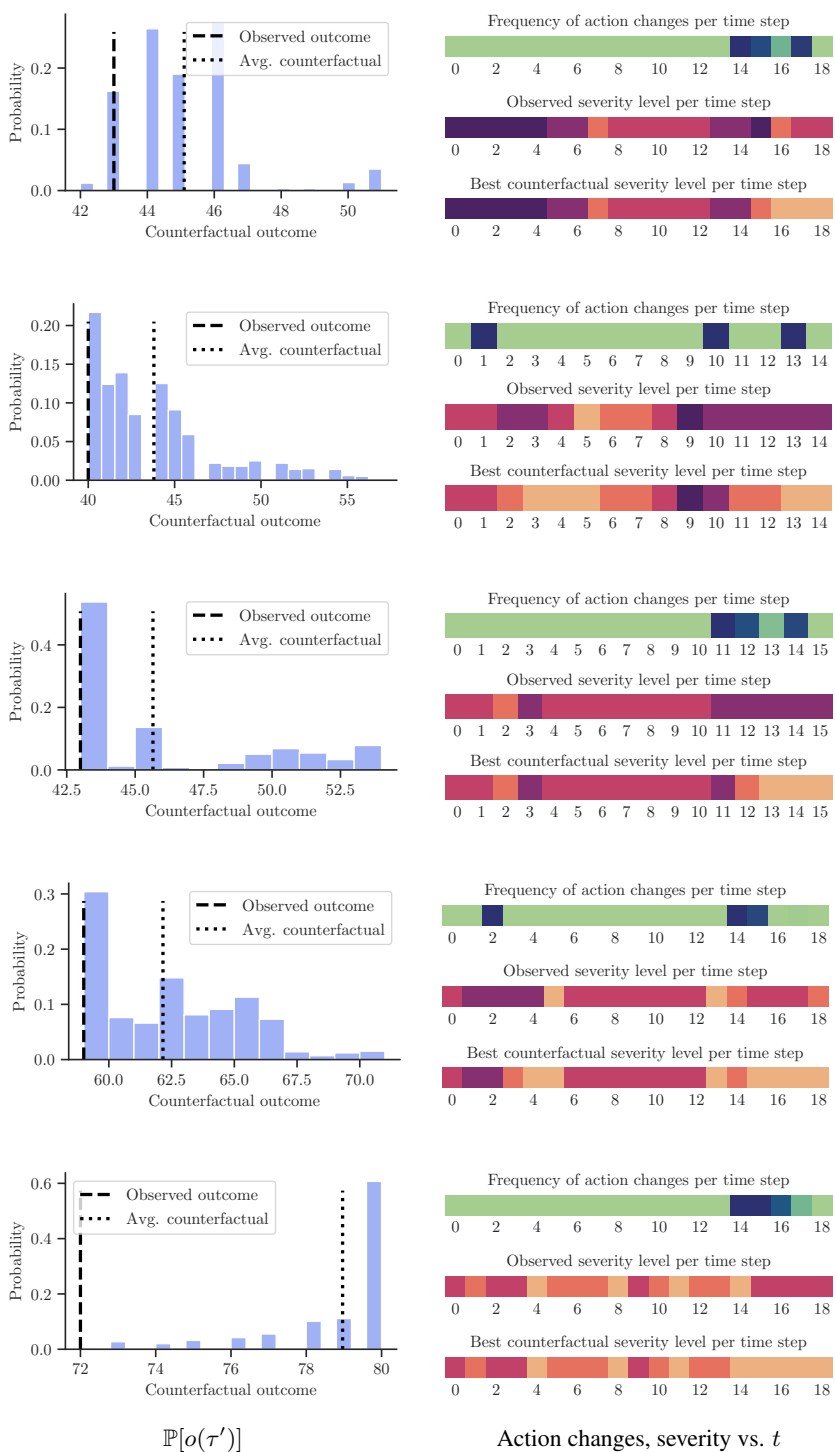

$\mathbb{P}[o(\tau')]$                    Action changes, severity vs. $t$

Figure 8: Insights on the counterfactual explanations provided by the optimal counterfactual policy $\pi_\tau^*$ for five real patients who received manualized cognitive behavioral therapy. Each row corresponds to a different patient with an observed realization $\tau$. The panels in the left column show the distribution of the counterfactual outcomes $o(\tau')$ for the counterfactual realizations $\tau'$ induced by $\pi_\tau^*$ and $P_\tau$. The panels in the right column show, for each time step, how frequently a counterfactual explanation changes the observed action as well as the observed severity level and the severity level in the counterfactual realization with the highest counterfactual outcome. Here, darker colors correspond to higher frequencies and higher severities. In all panels, we set $d = 1,000$, $k = 3$, and the results are estimated using 1,000 counterfactual realizations.