# OpenReview forum: "Counterfactual Explanations in Sequential Decision Making Under Uncertainty"
_NeurIPS.cc/2021/Conference — NeurIPS 2021 Poster_

### Official Review · Reviewer_bT61 · 2021-07-07

**Rating:** 7
**Confidence:** 4

**Summary:**

This paper is about counterfactual explanations for Markov decision processes. Instead of the usual counterfactual explanations as "alternative traces which can be used to explain that the chosen (optimal) choices are in fact optimal" as in explainable AI and reinforcement learning (RL), this paper uses them in offline settings, where data is available of the chosen paths, and where potentially better alternatives (counterfactual) can be computed. The main idea is to employ structural causal models to capture the model of the MDP such that better alternatives (here: action sequences that differ at most in k actions) can be computed in that model. The paper features experiments in two domains; one artifical and one in a small (in terms of the model) medical treatment domain. The experiments clearly show the benefits and characteristics of the approach.

**Ethical Concerns:**

None.

**Limitations And Societal Impact:**

Not discussed (limitations; the paper merely lists future work but not addresses the limitations explicitly) and the societal benefits/impact is seen in the contribution to interpretable machine learning. I think that the latter point is fair and does not require further elaboration than is now in the paper.

**Main Review:**

Overall this is a nice and fairly well-written paper. It is clear in what it does, the formalization and algorithms make sense, and the experiments are insightful and effective. Downsides are a limited treatment of related work and connections to other work, some doubts about the relative novelty/significance, and some missed opportunities to structure the conceptual story in a better way. I do not think the specific modeling choices are well motivated nor put well in the context of the literature. I will elaborate.

Section 1 is informative and clear, but I wish the authors would have had a better grasp of the surroundings, and had provided a better story to situate the work. On one hand this is about the build-up of the motivation. So far, the story is "counterfactual explanations can be done in single decisions and typical machine learning contexts" followed by "and now we move to sequential decisions". This is also reflected in the related work. However, counterfactual explanations are typically used in a different way as in this paper: namely to show that alternative paths are not as optimal as the chosen one, instead of showing that alternative paths are indeed better than what is chosen now (reflected in the data). Other work in explainable RL and planning exist where such typical meanings of "counterfactual explanations" are computed. That side of the story is not related to at all, also not acknowledging that there is a body of work on explainable AI for sequential decision making. The other point is about novelty. It seems that references such as [19] are used for some technical aspects, but in fact, [19] already does many (conceptual) things as in this paper, such as the structural causal models, counterfactual explanations, but then even for POMDPs. A clear separation between this paper and other ones using these kinds of models is needed to clearly see the contributions of the paper, and is lacking now in the paper.

The storyline in sections 2-4 sometimes looks quite difficult but can be (and should be) summarized much more clearly. As far as I can see, the paper takes offline data of some decision making process (traces) and uses that to estimate a model of an MDP. This is not an easy task (identifiability) hence structured causal models are employed, and some sampling mechanisms in there, but the bottom line is that from that point on, there is a model which can be used to find better traces (better action selection). In order to do that structuredly, the authors look at "at most k differences" in alternative traces. The construction of the extended MDP (with how many alternatives left) and the corresponding dynamic programming frame are straightforward then, but the specifics not much relevant (alternatives exists, since this is all based on a fully specified MDP at that point). This clear conceptual story (I'm assuming I read it right, of course) is lacking and could make the paper much better. Also, some better explanation/illustration of the structured causal models (as in [19]) could help a lot. Some of the formalization of the probabilities is not concise, and layout-wise formulas such as in (3) can be structured much better and clearly. The "do-notation" should be formalized better here too. The stability aspects at the end of page 3 can be illustrated better, since these are quite important because they determine the possibility of learning the model.

A main question that is not asked in this paper, and would be highly relevant also for interpreting the results is: how good is the model that is estimated from data? The better the model, the more credible are statements about the relative quality of solutions (and thus the quality of counterfactual explanations). Not much is said about this, even though it is crucial for everything. This should be answered in some way.

Another question that is not asked in the paper (related to the previous) is how well the counterfactual explanations would do in the real world. I mean, the model is an approximation of the real world, with assumptions to make things work. What would happen if it would be possible to really apply them, and how would the outcomes differ? The first experiment could give some insight here, but it is not easy to see whether the setup is too clean/regular or easy (for the model approximation) for that?

I like the experiments. They mostly make sense and clearly shows the main dimensions of the approach. Fig 1 is clear and logical. Fig 2 is mostly ok, but (c) and the description in the text raises some questions since it is quite a binary thing: either alpha is 0, or it is larger than 0. If alpha is only 0.1 we already get the same kind of pattern as with alpha = 1.0. This is not well explained yet. Also the sentence "however.... increases" (232/233) is not clear to me. The second experiment is called "real data" but the underlying MDP is very tiny, which asks for a good description of how particular aspects of this dataset (that is is from real treatments, how many trials, how long the trials are, and how large or small the underlying MDP is (in this case 5x9=45 state-action pairs max)). These questions are triggered since the "artificial" problem in the first experiment is larger (the MDP size) but more regular possibly. For Fig 3 the results make sense and the corresponding text too, but the choice for k=3 is not motivated (although intuitive maybe given the small MDP size, but on the other hand the total length of traces is about 20, so why not k=10?). Fig 4a is not clear to me, but 4b and 4c are, and I think that the analysis of this particular case is nice, although I wish a more comprehensive analysis would be done on more traces (this one feels illustrative, but also somewhat anecdotical).

There some small typos such as "a MDP" -> "an MDP" (end of page 2). Some references are made to Algorithm 2, but sometimes it is not clear it is in an appendix.

POST-REBUTTAL: thank you for your adequate answers and promises. I will keep my score.

**Time Spent Reviewing:**

4

---

> ### Author Response · Authors · 2021-08-06
> **Initial author response**
>
> We would like to thank the reviewer for insightful comments and suggestions, which will help improve our paper.
>
> To contextualize our work, we have compared our work to previous work on counterfactual explanations for classification. This is because, in our literature search, we did not find references to "counterfactual explanations" in explainable RL and planning, except for one notable exception [14]. However, following the reviewer's advice, we will better contextualize our work with respect to explainable RL and planning and differentiate our work from [14] more clearly. We would be very thankful if the reviewer could kindly provide additional relevant citations.
>
> We recognize that the model we used in our work builds upon [19], as explicitly noted in lines 53-60 and 116-117. However, as discussed in lines 51-53, [19] solves a different problem. It introduces an off-policy evaluation procedure for highlighting specific realizations of a sequential decision making process where a policy trained using RL would have led to a substantially different outcome. In our work, given a specific observed realization of a sequential decision making process, we find alternative action sequences that are "close" to the observed one that could have led to better outcomes. We will expand our discussion of [19] to better differentiate the contributions.
>
> We will improve the writing of the storyline of our paper so that it is conceptually more clear. In that context, we would like to clarify one aspect -- maybe the reviewer already understood this aspect but it is unclear to us from his/her review. We use the observed traces to fit the transition probability of the MDP and then construct an extended MDP to facilitate the search over policies that satisfy the constraint "at most $k$ differences". However, for each given observed trace, we do find a different counterfactual policy using our algorithm. And, to run our algorithm, we need to estimate the posterior distribution of the Gumbel-max noise given the observed trace and use it to estimate the counterfactual probability distribution.
>
> We will include an illustration of the SCM, make the formalization of the probabilities more concise, improve the layout of the formulas and the formalization of the "do-notation", and expand on the stability aspects.
>
> We agree that the usefulness of the counterfactual explanations to a decision maker crucially depends on how good the model we use is, i.e., how good the transition probabilities $P(s' | s, a)$ are at characterizing the environment dynamics and how good the SCM is at characterizing the counterfactual environment dynamics. Regarding the transition probabilities, our methodology does not depend on the choice of its functional form---one could use any sophisticated model (e.g., a neural network). Regarding the SCM, since one cannot really observe the counterfactual environment dynamics, the choice of SCM is instead motivated by a desirable stability condition (refer to [19] for a discussion of this condition). However, our DP algorithm is agnostic to the choice of SCM and more precise domain-specific SCMs could be used in practice. We will discuss this in the revised paper.
>
> If our methodology would be deployed in practice, the counterfactual explanations would not be "applied" since they refer to past realizations of the decision making process that already happened. The counterfactual explanations would be used to provide insights to a human decision maker about what could have happened if they would have acted differently in a specific past situation. Given these insights, the decision maker might or might not change their action policy in the future.
>
> In the synthetic experiments, we will rephrase the sentence in lines 232-233 and regenerate Fig 2(c) with more values of $\alpha \in [0, 0.1]$. Regarding the latter, please note that, for $\alpha=0.1$, the number of distinct counterfactual explanations is lower than for $\alpha = 1$.
>
> Due to space constraints, we deferred most of the details about the dataset to Appendix C. If our paper is accepted, we will move some of this additional content to the main using the allowed extra page for the camera-ready version and provide additional statistics about the dataset.
>
> We only show the average counterfactual improvement for $k=3$ in Fig 3a and insights about one single trace in Fig 4 also due to space constraints. If accepted, we will use the allowed extra page to show results for other values of k and move insights about additional traces currently in Fig 6 in the Appendix to the main.
>
> We will include a standalone section stating the limitations, which are currently discussed separately. It will acknowledge that our method only allows discrete states and actions and it assumes that the sequential decision process satisfies the Markov property, and that our validation uses only one real dataset.
>
> We will proofread our paper to correct any typo and explicitly write that Algorithm 2 is in the Appendix.

---

> > ### Comment · Reviewer_bT61 · 2021-08-19
> > **about counterfactuals and RL/planning**
> >
> > There's an earlier tutorial in this direction (https://cs.stanford.edu/people/ebrun/Brunskill_RLDM_Tutorial_2019.pdf), I think "Generating Explanations for Temporal Logic Planner Decisions by Kasenberg et al (2020)" is (although a very different setting) a good source for further exploration, and I know I have reviewed other work in the past and while searching from memory I just found this very recent survey that provides a good general source for work on these aspects (https://ieeexplore.ieee.org/document/9321372)

---

> > > ### Author Response · Authors · 2021-08-20
> > > **Many thanks for providing citations**
> > >
> > > We thank the reviewer for providing useful citations at the intersection of RL, explainability and counterfactuals. We will cite the aforementioned survey and we will use resources from the proposed tutorial and paper to enrich our discussion on related work and better contextualize our paper.

---

### Official Review · Reviewer_vTQ9 · 2021-07-11

**Rating:** 4
**Confidence:** 3

**Summary:**

This paper proposes a novel technique to generate counterfactual explanations for Markov decision processes (MDPs). Technically, it first remodels the state transition of MDPs with a Gumbel-Max structural causal model. Then, it formalizes the counterfactual explanation generation as an optimization problem and proposes an algorithm to generate the explanations. With the experiments on both a synthetic and a real-world dataset, the paper demonstrates the effectiveness of the proposed technique in finetuning a given policy to collect more rewards, especially when the corresponding environments have uncertainties.


**Limitations And Societal Impact:**

IMHO, this paper neither sufficiently discusses the limitations nor the negative social impacts. Regarding limitations, the authors could refer to my above comments. For the potential negative societal impact, they are only briefly discussed in the checklist. More strictly, the discussion in checklist 1(c) is actually not about the "negative" social impact. I would suggest the authors include such a section in the paper or appendix to discuss the negative impact of their proposed technique. For example, as an explanation tool, it can be used by attackers to generate attacks against MDPs or a target policy.

**Main Review:**

This paper extends existing counterfactual explanations on DNN to MDP and generates explanations for target policies acting in the corresponding MDPs. First of all, I like the idea of generating counterfactual explanations for MDPs. This explanation method takes into account the sequential decision process and can be potentially used to identify the important actions/time steps in a trajectory of a target policy. In the context of DRL explanations, most of the existing research focuses on explaining the neural policies or the Q networks. Not too many works tried to explain the whole decision-making process. Although the problem studied in this paper is novel, the following concerns pause me from supporting it.

1. I found the paper is relatively hard to follow, especially the technical part. The reasons come from three folds: (1) The paper does not provide a sufficient introduction of the necessary backgrounds. For example, it would be helpful if the authors could quickly introduce and summarize the SCM used in this work. The arguments in this work largely depend on [18] and [19]. It would be easier for readers to understand these arguments if the authors could review those two works and summarize their key points. These backgrounds can go to the appendix. For readers that are not familiar with causal influence and causal-based explanations, they are super-helpful.  (2) Some important notations and formulations are not very clearly explained (See the below comments for more details). (3) This work frequently uses long and complicated sentences. For example, lines 103-107 are one sentence. To improve the readability, I would highly suggest rewriting those sentences with simpler structures. Overall, I found it is relatively hard for me to understand every component of the proposed technique because of the above reasons. I can only get the high-level idea is that given a trajectory of a target policy, the proposed technique seeks to search for the best K actions to change, keep the other actions, and increase the collected rewards.

2. Here, I list the places where I found it is hard to understand. They may be caused by the reasons I mentioned above or my lack of relative knowledge. IMHO, addressing these comments could help readers better understand the technique proposed in this paper. I would really appreciate it if the authors could do so.

(1) The motivation of using an SCM to remodel an MDP is unclear. In the sentence starting from line 89, the authors argue the weakness of MDP is that MDP is not able to tell what would happen if changing one action $a_t$ in a trajectory. I found this argument is relatively vague. "What would happen" is an especially unclear description. From my point of view, what would happen is starting from the time step $t+1$, the agent will jump into a different state from the previous one, and the following actions/states will change correspondingly. The changed trajectory can be obtained by running the target agent in the corresponding MDP (repeating the trajectory till the $t$-th step, taking a different action at that step, and continuing running the agent in the following steps). To this end, MDP could tell what would happen if changing $a_t$. The above argument is invalid. I guess what the authors want to express if MDP is not able to simulate the case where only $a_t$ is changed, and the following states/actions keep the same. According to the above discussion, in MDPs, changing one action will likely trigger the changes in the following states/actions. With the unclear limitation, I was also not able to understand the benefits of using SCM. To make it more clear, maybe the authors could consider reexpressing the limitation of MDP and clarifying what is the characterization in line 93 and what are the desirable properties in line 94.

(2) The posterior distribution in line 104 is not clearly presented. First, what does it represent? My understanding is it defines a distribution of $U_t$. If the agent takes $a_t$ at $s_t$, sampling any $u_t$ from this distribution can transit the system from $s_t$ to $s_{t+1}$. Second, It would be easier to understand if the authors could demo one such distribution. My understand is the distribution should be 1[g_s(s_t, a_t, u_t) = s_{t+1}]. If this is the case, how to compute the density function. (3) What is the prior and what is the likelihood.

(3) Without a clear understanding of the above posterior, it is hard for me to fully understand Enq. (3). I would appreciate it if the authors could provide more explanations of the first step. First, what does the left-hand side of the equation stands for? Second, the right-hand side looks very similar to the right-hand side of Eqn.(2). What is the difference? Some text descriptions of those two terms are needed. They are important and deserve more explanations.

(4) Regarding the non-identifiable problem, the paper refers to the argument [19]. As is discussed above, it would be helpful if the authors could summarize what the argument in [19] is.

(5) The condition after 122 implies the equation in line 126, and it also implies the conditions after line 125. However, since the conditions after line 125 cannot infer the condition after line 122, it cannot imply the equation in 126.

(6) What is the connection between the model and the condition?

(7) Algorithm 1 is redundant, It just expresses a normal testing process. I would highly suggest explaining more about Algorithm 2. With all the above questions, I found Algorithm 2 very hard to understand.

3. Comments about the evaluation:

(1) It worth clarifying what an explanation is. My understanding is that an explanation refers to a trajectory with counterfactual actions.

(2) I would suggest reporting the runtime of the proposed method.

(3) It is important to study the boundary and failure cases of a proposed technique. I am wondering if there are cases where the proposed method fails to generate explanations.

(4) I would like to see the applications of the proposed method to more complicated environments with more discrete actions and states.

4. Finally, the paper does not discuss the possible limitations and how to solve them.  Below, I point out some possible limitations of the proposed technique that worth discussing.

(1) Scalability. As is mentioned above, the environments evaluated in this paper are relatively simple ones with a small number of states and actions. It worth discussing the scalability of the proposed technique to more complicated environments.

(2) Practicability is another major concern. The proposed technique is limited to environments with discrete actions and states. If the authors are not able to demonstrate the scalability of the proposed technique, then its practicability will also be largely restricted.

(3) As is mentioned above, it is worth evaluating and discussing the boundary of a new technique. For the proposed method, I am wondering if it still works if the target policy is an optimal one. In addition, I assume the proposed technique may fail if the game has only a final reward without an instant reward.  I would suggest discussing these cases.

(4) Existing counterfactual explanations of DNN are able to identify the most important input features, modifying which may cause the target model to make a mistake. The proposed technique aims to improve the target policy's performance via action modification. I am wondering if it can operate in the opposite way, i.e., modifying actions to trigger the highest reward deduction.


=========Post rebuttal=============

Thanks to the authors for providing the rebuttal. It indeed addresses some of my concerns. Regarding that, I will not be strongly against accepting this work if the other reviewers and AC all vote for an acceptance. IMHO, the paper still has room for improvement. Below, I listed some comments that the authors could consider addressing if the paper got accepted (The authors do not need to reply to the following comments at this moment given the discussion is coming to an end. They are (non-mandatory) suggestions for camera-ready if the paper got accepted).

**Clarity and readability.** I read the responses together with the paper multiple times. Despite having a better understanding of the overall procedure of the proposed technique, unfortunately, I was still not able to understand most of the technical details. This may be due to my lack of knowledge in causal inference and causal model (My knowledge about causal inference is at a relatively high level - understand the definition of the casual model, intervention, and counterfactual. But I do not have a detailed grasp of specific casual models, such as the one used in this work.). Maybe for casual inference experts, the current version is enough for them to understand more details. However, for general audiences, the current descriptions of the proposed technique are not that clear. Specifically, it is quite intuitive that $P(S_{t+1}=s_{t+1}|S_{t}=s_{t}, A_{t}=a_{t})$ with/without the corresponding trajectory. From this fact, it seems natural to infer that $P(S_{t+1}=s_{t+1}|S_{t}=s_{t}, A_{t}=a')$ would be also different. However, it is still not entirely clear to me what's the difference between the original state transition function and the intervention/counterfactual distribution. Eqn. (3) deserves more space to explain each line of derivation. The non-identification problem is more clear with the response. Line 122-126 has logical errors (as I pointed out in my review). I would suggest correcting them. If the posterior in line 105 is different from the general definition, it should definitely be pointed out. Otherwise, it will cause misunderstanding. In Sec. 3, the authors define another model - the counterfactual MDP model, with a new set of state transitions. I would suggest emphasizing their difference from the ones defined in Sec. 2, especially a text description of Eqn. (6) would be very helpful. The last point that can be more clear is Algorithm 2. IMHO, Algorithm 2 is much more important than Algorithm 1, given that it is the one used to generate explanations. I read the current version in the Appendix. Again, I can only get a high-level idea - recursively search for a better action at each step within the budget K. I would suggest moving Algorithm 2 to the main text and add a line-by-line explanation to the Appendix to help readers better understand it. Overall, IMHO, the current version is enough for the general audience to understand the proposed technique at a high level. However, to make the technical details more clear and accurate, the paper may require some major revisions, as I suggested earlier.

**Faithfulness of SCM.**
As the authors mentioned in the response, the reason for using SCM is to model what would have happened if manipulating some actions in a given trajectory of an agent. The example of the current state and action given by the authors makes sense to me. However, this example may not be enough to justify that MDP cannot capture what would have happened if the actions in a given trajectory are changed. A counterargument is MDP reflects the state transition of the original system. If an existing action changes, the new action would be given to the system, and the system transits according to the transition function of the MDP. I would appreciate it that if the authors could explain more about the difference between normal state transition ($P(S_{t+1}| S_{t}, A_{t})$) and the state transition when changing the actions of a given trajectory ($P_{\tau}(S_{t+1}| S_{t}, A_{t})$). My point is as long as the action is acted in the same environment, they should be the same. Note that this is different from the example given in the authors' responses, where no action changing is involved (P(S_{t+1}=s_{t+1}| S_{t}=s_{t}, A_{t}=a_{t})). Of course, my understanding may be wrong. Despite $P(S_{t+1}| S_{t}, A_{t})$ and $P_{\tau}(S_{t+1}| S_{t}, A_{t})$ are different, my second question is why SCM could model the $P_{\tau}(S_{t+1}| S_{t}, A_{t})$. This is where more backgrounds of SCM are needed. In SCM, there is a noise term. I would doubt that whether introducing this noise term could still represent the original system. The same question can also be raised for the counterfactual MDP model. Those questions make me doubt the accuracy of the model (the same question pointed out by the reviewer bT61). The author didn't directly respond to that question with an evaluation. These concerns may still stand.

**Explainability of the proposed technique.**
To me, the proposed method is more like a policy fine-tuning approach rather than explanation methods. The proposed method generates a set of counterfactual trajectories that have higher rewards than the original ones. By comparing these trajectories with the original ones, some insights about the original policy can be drawn. The explainability of the proposed approach was qualitatively demonstrated in some simple environments. To further justify the explainability, the authors may need to design and conduct some quantitive evaluation or human studies, just like most of the existing DRL explanation works have done (e.g., [1], [2], [3]).

**Scalability, Practiabilty, and generalizability.**
Another major concern is about Scalability, Practiabilty, and generalizability. Regarding scalability, the computational cost is quadratic to the number of states. This significantly limits the applications of the proposed technique. In the evaluation, the authors only tried a real-world dataset with $4$ states. Besides, the proposed technique can only be applied to systems with discrete state space. I would highly doubt the practicability of the proposed technique to environments that are widely used in the DRL (explanation) community - openAI gym [1], Atari games [1,3], Go game [2], etc. and derive useful insights. This is a fundamental weakness that may not be addressed through paper revising or additional experiments.

Relatively minor points: 1. a comprehensive literature review of the existing DRL explanation techniques is highly encouraged. 2. In the responses, the authors state that "Finally, there are no cases where our method “fails” to generate counterfactual explanations. The only interesting edge cases are the ones where there is no action change which, in retrospect, could have given an average counterfactual outcome higher than the observed one". I would like to see how often those cases will happen. If the target agent has an optimal policy, It may be quite frequent that the proposed technique cannot find better actions that would have given a higher reward.

[1] Visualizing and understanding atari agents.

[2] Explain your move: understand agent actions using specific and relevant feature attribution.

[3] Exploratory Not Explanatory: Counterfactual Analysis of Saliency Maps for Deep Reinforcement Learning.

**Time Spent Reviewing:**

7 hours

---

> ### Author Response · Authors · 2021-08-06
> **Initial author response**
>
> We would like to thank the reviewer for the insightful comments and suggestions, which will help improve our paper. Throughout our response, we do explicitly refer to each of the reviewer's points X as "[Re. X]".
>
> Since the reviewer has raised several questions regarding our problem definition and the need to use structural causal models (SCMs), or causal modeling more generally, to solve our problem, in the next 3 paragraphs, we would like to provide some general background material to better motivate the need to use causal modeling to define and solve our problem -- maybe the reviewer already knows this background material but it is unclear to us from his/her review.
>
> **[Re. 1 & 3.1]**
> We assume the existence of a sequential decision making process which can be characterized by a Markov Decision Process (MDP). Then, we aim to generate counterfactual explanations for observed realizations (or in other words, observed trajectories) of the process (i.e., action-state sequences that have already happened), without any knowledge about the action policy that generated the actions. By “counterfactual explanation”, we mean an alternative action sequence differing from the observed one in at most $k$ actions that, **in retrospect**, could have given a higher reward. We are being very precise with the use of the phrase “could have given” instead of “could give” because counterfactual explanations are referring to the past. This explains why, in addition to characterizing the process using an MDP, we need to additionally characterize the process using an SCM, a point that we discuss in detail in the following paragraph.
>
> **[Re 2.1]**
> Assume that we observe a decision making process is **currently** at state $s _t$, at time $t$, and we would like to answer the following question: “What is the probability that, if we take action $a _t$, the next state is $s _{t+1}$?” This question can, indeed, be answered by a simple MDP, without using an SCM and the answer is $P(S _{t+1}=s _{t+1} \thinspace | \thinspace S _t=s _t, A _t=a _t) \leq 1$.
>
> Next, assume that we have observed the entire realization of the process $\tau = \{(s _t, a _t)\} _{t=1}^{T}$ until its end at time $T$. Then, we know that, at time $t$, it transitioned from $s _t$ to $s _{t+1}$ after using action $a _t$. Now, we would like to answer the following question: “What is the probability that, if we had taken action $a _t$ at time $t$, the next state would have been $s _{t+1}$?” Since this question is asking something about the past that we have already observed, the probability is not $P(S _{t+1}=s _{t+1} \thinspace | \thinspace S _t=s _t, A _t=a _t)$ anymore but, instead, it is strictly equal to $1$. Following the same reasoning, the answer to the (counterfactual) question “What is the probability that, if we had taken a different action $a’$ at time $t$, the next state would have been $s _{t+1}$?” is not necessarily equal to $P(S _{t+1}=s _{t+1} \thinspace | \thinspace S _t=s _t, A _t=a’)$. To answer this type of counterfactual questions about the past, we need to use an SCM, as discussed next.
>
> **[Re. 2.1, 2.2 & 2.3]**
> In an SCM $\mathcal{C}$, a (random) transition between states is expressed by means of the assignment $S _{t+1 }= g _S(S _t, A _t, \mathbf{U} _t)$ (see Eq. 1), where $\mathbf{U} _t$ is a noise variable. The distribution $P^{\mathcal{C} \thinspace ; \thinspace do(A _t=a _t)}(S _{t+1} \thinspace | \thinspace S _t=s _t)$, entailed by such an assignment, is what we call the **interventional** distribution. Moreover, for Gumbel-Max SCMs, where $\mathbf{U} _t$ takes values from a Gumbel distribution, this interventional distribution is the same as the conditional transition distribution $P(S _{t+1}=s _{t+1} \thinspace | \thinspace S _t=s _t, A _t=a _t)$ of a simple MDP, as noted in Eq. 2.
>
> In this context, the distribution $P^{\mathcal{C} \thinspace | \thinspace S _t=s _t, S _{t+1}=s _{t+1}, A _t=a _t \thinspace ; \thinspace do(A _t=a)}(S _{t+1} \thinspace | \thinspace S _t=s)$ is called the **counterfactual** distribution and, for simplicity, we refer to it as $P _{\tau, t}(S _{t+1} \thinspace | \thinspace S _t=s, A _t=a)$ (see first line of Eq. 3) because it depends on the observed transition at time $t$ in the realization $\tau$. Here, $\mathbf{U} _t$ does not follow a Gumbel distribution anymore but, instead, it takes values $\mathbf{u} _t$ from a **posterior** distribution $P^{\mathcal{C} \thinspace | \thinspace S _t=s _t, S _{t+1}=s _{t+1}, A _t=a _t}(\mathbf{U} _t)$ with support such that $\mathbb{1}[s’=g _S(s,a,\mathbf{u} _t)]$. Note that this differs from the notion of the posterior being proportional to the product **likelihood** $\times$ **prior**, as it appears in Bayesian inference. In our work, we do not compute its density $f^{\mathcal{C} \thinspace | \thinspace S _t=s _t, S _{t+1}=s _{t+1}, A _t=a _t} _{\mathbf{U} _t}(\mathbf{u})$ analytically but we sample from it (see lines 127-129) to estimate the counterfactual transition distributions (see Eqs. 3,5).
>
> **[Re. 2.4, 2.5 & 2.6]**
> Regarding the identifiability of the underlying SCM, the argument is that, there could be multiple combinations of functions $g _S$ and distributions for $U _t$ which give interventional distributions $P^{\mathcal{C} \thinspace ; \thinspace do(A _t=a _t)}(S _{t+1} \thinspace | \thinspace S _t=s _t)$ consistent with the MDP’s transition probabilities but result in different counterfactual transition distributions $P^{\mathcal{C} \thinspace | \thinspace S _t=s _t, S _{t+1}=s _{t+1}, A _t=a _t \thinspace ; \thinspace do(A _t=a)}(S _{t+1} \thinspace | \thinspace S _t=s)$. Since this is not an observation originally made in our work, we briefly discuss it in lines 112-116 and we refer to [19] for a further discussion. Therefore, we restrict our focus to the Gumbel-Max SCM, which satisfies a natural counterfactual stability property (explained in lines 119-126). Regarding the property, it is formally proven in [19] and, for completeness, we give an intuitive explanation of it in lines 121-124 and a formal statement in 125-126. Note that the condition after line 125 is more general than the one given after line 122, which is used solely to make the explanation of the condition more intuitive.
>
> **[Re. 3.3 & 4.3]**
> We would like to clarify that our method may return an average counterfactual outcome higher than the observed one even if the policy followed to generate the observed trajectory is optimal. This is because, when looking into the past, the actions provided by the optimal policy may not be optimal since, in retrospect, we have additional information about the realized transitions (i.e., information about the noise posterior). Moreover, we would like to mention that the case where the MDP gives only a final reward, without instant rewards in immediate states (those can be equal to 0), is a special case where our method still works. Finally, there are no cases where our method “fails” to generate counterfactual explanations. The only interesting edge cases are the ones where there is no action change which, in retrospect, could have given an average counterfactual outcome higher than the observed one. However, in those cases, as we discuss in lines 160-161 and 220-224, our method would trivially return a single counterfactual explanation (the observed action sequence) which would give a counterfactual outcome equal to the observed outcome with probability 1.
>
> **[Re. 3.4, 4.1 & 4.2]**
> We believe that our formulation fits a variety of real world scenarios where the decision making process consists of discrete states and actions (like the one appearing in our experiments on cognitive behavioral therapy data) and a decision maker makes sequential decisions based on a small set of available actions, as we discuss in lines 43-46. However, extending our approach to continuous state-action spaces is an interesting direction for future work, as we mention in lines 328-329.
>
> **[Re. 2.7, 3.2, 4.4 & comment on Limitation and Social Impact]**
> If our paper gets accepted, we will use the extra page of the camera-ready to include a standalone section clearly discussing the limitations of our approach, we will bring Algorithm 2 to the main and, we will include an additional plot about runtime, following the reviewer’s suggestions. Moreover, we will include more details about the SCM, its identifiability and related background from [18, 19] in the Appendix to make the paper accessible to a larger set of readers, who might not have a background on causal modeling.
>
> We hope our response proves itself useful for the reviewer to fully understand the details of our work and evaluate the significance of the proposed method. In light of the above clarifications, we would like to kindly ask the reviewer to reconsider their score and we are at their disposal for any further questions.

---

### Official Review · Reviewer_161w · 2021-07-16

**Rating:** 8
**Confidence:** 4

**Summary:**

The paper proposed a novel method to generate counterfactual explanations. The authors view the problem as sequential decision-making processes using Markov decision processes and the Gumbel-Max structural causal model. They discuss an algorithm based on dynamic programming to find optimal CFEs. The model was validated on synthetic and cognitive behavioural therapy datasets.


**Limitations And Societal Impact:**

Limitations are not sufficiently discussed.

**Main Review:**

Originality:
This is an exciting piece of work. The idea of using an MDP to do something similar has been introduced recently elsewhere, but the model introduced in this paper is novel.

A slightly related field I think is automated planning and learning. See for example:

Jiménez, S., De La Rosa, T., Fernández, S., Fernández, F., & Borrajo, D. (2012). A review of machine learning for automated planning. The Knowledge Engineering Review, 27(4), 433-467.

Quality:
Overall, the paper appears to be technically sound. However, I think the evaluation needs to improve in two ways:

1) As with standard ML-based approaches, there should be a training and test set. This will allow the authors to evaluate the effectiveness of the approach better.

2) Comparison with existing methods: I understand the conceptual difference between existing methods and the model proposed in this paper. However, I feel that there is nothing stopping people from taking sequential action of the existing method suggests changing multiple features -- I could change each feature one at a time.

Clarity:
The paper was well organised, the concepts easy to follow, and the results were explained reasonably well.

Significance:
This work is important to the algorithmic recourse community and has the potential to open a very interesting and promising area of research.

**Time Spent Reviewing:**

1-2

---

> ### Author Response · Authors · 2021-08-06
> **Initial author response**
>
> We would like to thank the reviewer for the insightful comments and suggestions, which will help improve our paper.
>
> We will contextualize our work with respect to automated planning and learning and cite the above mentioned review, following the reviewer's suggestion.
>
> Please, note that, in our experiments, the goal is not to evaluate the accuracy of the model of the transition dynamics of the MDP and the Gumbel-max structure causal model. Instead, our goal is to evaluate the counterfactual outcomes achieved by the counterfactual explanations found by our method given the model and the observed realizations. To evaluate the accuracy, we agree that one would typically create a training and test set as the reviewer suggests. However, we cannot think of any way in which one could use a training and test set to evaluate the counterfactual outcome achieved by the counterfactual explanations.
>
> Due to space constraints, we deferred a comparison with several competitive baselines to Appendix D. In that context, please note that, at each step, the greedy and the noisy greedy baselines find alternative actions of immediate maximum reward and therefore one could see those alternative actions as counterfactual explanations of minimum cost in classification [7, 8]. If our paper is accepted, we will move the comparison with the baselines to the main using the allowed extra page for the camera-ready version.
>
> We will include a standalone section stating the limitations, which are currently discussed separately throughout the paper. More specifically, we will acknowledge that our method only allows discrete states and actions and it assumes that the sequential decision process satisfies the Markov property, and our validation uses only one real dataset.

---

### Official Review · Reviewer_QEFf · 2021-07-17

**Rating:** 6
**Confidence:** 3

**Summary:**

The paper proposes a method using Markov decision process and dynamic programming to build alternative sequences with better reward/ outcome. The algorithms used can find optimal counterfactual explanations in polynomial time, testing in both synthetic and real dataset. With different uncertainty (defined manually), optimal counterfactual sequence outperformed the observed one considering the outcome in most cases. Thus the paper validated the contribution in finding the optimal counterfactual sequence in multiple, dependent sequence under discrete and low-dimensional situation.

**Limitations And Societal Impact:**

The paper mentioned the limitations: only discrete states and actions used; only one real dataset validated; manually decide the transition probability; limit the process to satisfy Markov property.
The paper fails to mention its potential negative societal impact. However, it is more of a theoretical research and thus few societal negativities can be concerned.
Experimental details. Please report the details required to reproduce the experiments. It would be more convinced if a GitHub repo is cited.



**Main Review:**

Originality:
The idea of applying counterfactual explanations in decision making process with multiple, dependent actions taken sequentially overtime is referred as original.
The algorithm is not replicated comparing to papers with the same topics.

Quality:
The paper is well-structured, it has solid research foundations based on previous work and has some originality concerning the whole experiment procedure. The inductions are very clear, though the dataset used is somehow lack of explanation. Provided more statistics in the experiments will certainly help to convince the audience.

Clarity:
The paper clearly describes its motivation, the method and experimental results. The method applied is introduced in an intuitive and clear way.

Significance:
The paper mainly contributes in illustrating the idea of finding optimal dependent sequences and validating the result in experiment. It is a solid practice of using Markov decision process, Gumbel-Max structural model in finding optimal decision sequence. Therefore, the idea and the experiment setting within this paper can be great reference for further work.


**Time Spent Reviewing:**

3

---

> ### Author Response · Authors · 2021-08-06
> **Initial author response**
>
> We would like to thank the reviewer for the insightful comments and suggestions, which will help improve our paper.
>
> Due to space constraints, we deferred most of the details about the dataset to Appendix C, a comparison with several competitive baselines to Appendix D and additional insights about the experiments to Appendix E. If our paper is accepted, we will move some of this additional content to the main using the allowed extra page for the camera-ready version and provide additional statistics about the dataset and experiments, following the reviewer's suggestion.
>
> To further facilitate reproducibility, we plan to release a public implementation of our method and the baselines on Github. To this end, we will build up on the code we included as supplementary material with our submission.

---

### Decision · Program_Chairs · 2021-09-27

**Decision:**

Accept (Poster)

**Comment:**

Based on the review, rating and discussion, I recommend acceptance of this work.

* there is no fundamental flaws in this work
* while the required background for reading the paper is heavy, the overall paper is well written.
* the significance of the contribution is high